# Discovery of fast and stable proton storage in bulk hexagonal molybdenum oxide

Tiezhu Xu[1], Zhenming Xu [1], Tengyu Yao[1], Miaoran Zhang[1], Duo Chen[1], Xiaogang Zhang [1] & Laifa Shen [1] ✉

Ionic and electronic transport in electrodes is crucial for electrochemical energy storage technology. To optimize the transport pathway of ions and electrons, electrode materials are minimized to nanometer-sized dimensions, leading to problems of volumetric performance, stability, cost, and pollution. Here we find that a bulk hexagonal molybdenum oxide with unconventional ion channels can store large amounts of protons at a high rate even if its particle size is tens of micrometers. The diffusion-free proton transport kinetics based on hydrogen bonding topochemistry is demonstrated in hexagonal molybdenum oxide whose proton conductivity is several orders of magnitude higher than traditional orthorhombic molybdenum oxide. In situ X-ray diffraction and theoretical calculation reveal that the structural self-optimization in the first discharge effectively promotes the reversible intercalation/de-intercalation of subsequent protons. The open crystal structure, suitable proton channels, and negligible volume strain enable rapid and stable proton transport and storage, resulting in extremely high volumetric capacitance (~1750 F cm$^{-3}$), excellent rate performance, and ultralong cycle life (>10,000 cycles). The discovery of unconventional materials and mechanisms that enable proton storage of micrometer-sized particles in seconds boosts the development of fast-charging energy storage systems and high-power practical applications.

The rapid development of economic society and growing demand for clean energy highlight the great importance of developing advanced energy storage technologies and materials[1–3]. Ideal electrode materials that can store more charge carriers at subminute time scales, with stable structure, are required to improve the energy, power, and safety of electrochemical energy storage[4,5]. Minimizing particle size of electrode materials to nanosized structure, a common method to improve the electrochemical performance of electrode materials, is used to enhance the charge transfer reaction on the electrode-electrolyte interface, to reduce the transport length of ions and electrons[6–8]. Some strategies based on electronic conduction-electrode nanocomposites and carbon-coating nanomaterials are also developed to optimize the transport path of ions and electrons, and diminish the distortion of the electrode structure[9,10]. Particularly, many advances in electrochemical energy storage technology (such as lithium-ion batteries[11,12], sodium-ion batteries[13,14], aqueous ion batteries[15], supercapacitors[16], etc.) have been achieved with the booming development of new nanometer-sized materials based on the advanced structure design and interface modification of electrode materials.

Unfortunately, larger volumetric space between nanometer-sized particles reduces the packing density of particles, resulting in a dramatically low volumetric capacity of electrode materials, which is not conducive to the fabrication and practical application of energy storage devices[17]. The initial Coulombic efficiency of nanostructured materials with high surface area is usually poor because of increased side reactions at the electrode/electrolyte interface. Furthermore, owing to the high cost, low yields, and potential environmental damage in the synthesis, characterization, and production of

[1]Jiangsu Key Laboratory of Electrochemical Energy Storage Technologies, College of Material Science and Technology, Nanjing University of Aeronautics and Astronautics, Nanjing 211106, People's Republic of China. ✉e-mail: lfshen@nuaa.edu.cn

nanomaterials, there are many difficulties in the commercialization of nanomaterials. In the next generation of energy storage systems, there is growing interest in creating micrometer-sized materials with fast charging time and long cyclic life for enhanced volumetric capacity, reduced side reactions, lower cost, etc. As an alternative to nanostructured electrode materials, two micrometer-sized $Nb_{16}W_5O_{55}$ and $Nb_{18}W_{16}O_{93}$ were reported, which delivered ultra-high volumetric capacity because of multielectron redox reaction and high packing density, charged and discharged at high current densities based on fast lithium-ion transport tunnels[6]. Kang et al. reported a phase separating material—submicron $LiVPO_4F$ particles with the formation of an intermediate phase during discharge, which also achieved ultrafast kinetics and high volumetric energy density[18].

Besides, as an important component of energy storage devices, charge carriers also determine the electrode reaction and material design[19]. Recently, due to the fast kinetics and high capacity of nonmetallic cations, many aqueous nonmetallic charge carriers batteries and supercapacitors (such as proton[20–23], ammonium-ion[24,25], etc.) have been developed, to address important problems such as poor power density, low safety, and high pollution in nonaqueous metallic ion batteries. Hydrous $RuO_2$, a typical surface pseudocapacitive material for proton storage, was found to achieve a high capacitance of over 700 F g$^{-1}$ in 1995[26]. $RuO_2 \cdot H_2O$ was regarded as a mixed electron-proton conductor, where the large-area nanocrystalline structure and rich structural water network provide fast transport paths for electrons and protons[27–29]. Many strategies to design nanostructures and confined water were developed to improve the electrochemical properties of $RuO_2$[30–32]. On the other hand, the high cost of ruthenium is pushing researchers to develop the next generation of novel proton-electron conductors for fast and stable proton storage. Recently, some typical electrode materials suitable for proton storage have been developed to bring insights into the renaissance of proton batteries/pseudocapacitors, including hydrated oxides[33,34], Prussian blue analogues[21,35,36], organic electrode materials[37,38], etc. However, all of the reported electrode materials (especially pseudocapacitive materials) for proton storage are in the nanoscale, leading to limited volumetric performance, serious side reactions, and complexity. There is no place for nanomaterials in commercial viability, such as manufacturing, safety, and cost[17]. It is critical to developing aqueous energy storage micrometer-scale bulk materials with fast chargeability and structural stability for large-scale energy storage.

Here we demonstrate that the suitable crystal structure design can endow bulk materials with an outstanding electrochemical performance at micrometer dimensions, which breaks the conventional nanostructuring strategy of electrode materials. We develop a micrometer-sized electrode material, bulk phase hexagonal molybdenum oxide ($h$-$MoO_3$), giving rise not only to an ultra-high volumetric capacitance (-1750 F cm$^{-3}$), but also to a stable electrode structure after thousands of cycles (10,000 charge-discharge cycles with negligible capacitance decay). We identify that the open three-dimensional channel structure, which enhances the ionic conductivity of $MoO_3$ by several orders of magnitude, provides a selective ion pathway to realize fast proton storage through the Grotthuss mechanism at an extremely high rate. Using in situ X-ray diffraction and density functional theory calculation, a structural enhancement process of $h$-$MoO_3$ is observed and reversible proton intercalation/deintercalation occurs during charge and discharge. A promising proton pseudocapacitor, which uses this $h$-$MoO_3$ electrode as the anode, achieves high specific capacitance, superior rate performance, and excellent cyclic stability.

## Results

### Structure and morphology characterization

$h$-$MoO_3$ is prepared by a simple hydrothermal method, and the synthesized sample is consistent with the X-ray diffraction (XRD) pattern of hexagonal $MoO_3$ in Fig. 1a, where sharp XRD peaks indicate an

$h$-$MoO_3$ with high crystallinity. In the Raman spectra, the strong peaks at 314, 396, 491, and 689 cm$^{-1}$ are attributed to the vibrations of O−Mo−O while the peaks at 886, 900, and 974 cm$^{-1}$ are caused by the stretching of Mo=O (Fig. 1b). The Fourier transform infrared (FTIR) spectra are also used to identify the structural characteristics of $h$-$MoO_3$, where the stretching vibration of O−Mo−O (500−700 cm$^{-1}$) and Mo=O (900–1000 cm$^{-1}$) is detected (Fig. 1c). Specifically, the apparent peaks associated with the stretching vibration (3000−3600 cm$^{-1}$) of O−H confirm that some water molecules are present in $h$-$MoO_3$. In contrast, the FTIR and Raman spectra of orthorhombic $MoO_3$ ($\alpha$-$MoO_3$) show remarkable change, indicating the different crystal arrangement of the $MoO_6$ octahedra in $\alpha$-$MoO_3$ (Supplementary Fig. 1a and b). The absence of the characteristic peaks for O−H illustrates the absence of structural water, which is attributed to the narrow lattice space in $\alpha$-$MoO_3$. According to previous literature[39,40], hexagonal $MoO_3$ has an open tunnel structure, in which large one-dimensional hexagonal tunnels (the internal channel diameter is ca. 0.94 nm) along the c-axis are formed through interconnecting $MoO_6$ octahedra (Fig. 1d). The open tunnel structure in $h$-$MoO_3$ provides fast electron and ion transport pathways to allow more ions to intercalate into or deintercalate from the host structure with lower diffusion energy, to enhance structural stability and fast kinetics of electrodes[41,42]. Distinguished from the confined interlayer spacing of layered $\alpha$-$MoO_3$ (Supplementary Fig. 2a and b), the $h$-$MoO_3$ even can give additional physical space to the confined fluids such as water molecules to assist ions to achieve a non-diffusion ion transport mechanism[33,34].

Figure 1e, f show typical scanning electron microscope (SEM) images of hexagonal microrods with a diameter of about 2−4 μm and a length of 10−30 μm, displaying a novel micrometer-sized structure. Thanks to ignorable void space between microparticles, the packing density of microparticle-based electrode materials tends to be high, to greatly increase the volumetric capacity and energy density of micrometer-sized materials. In contrast, $\alpha$-$MoO_3$ synthesized by a similar method shows a typical nanorod structure (Supplementary Fig. 3). Three edges around 1−2 μm of hexagonal microrods are directly observed in the transmission electron microscope (TEM) image (Fig. 1g), and a lattice spacing of 0.27 nm that corresponds to the (220) plane is found in the high-resolution TEM image (Fig. 1h). The element mapping image demonstrates the uniform distribution of Mo and O element (Fig. 1i). Thermogravimetric analysis (TGA) is used to determine the water content in $h$-$MoO_3$ (Supplementary Fig. 4), in which three different stages of mass loss are found: a) the mass loss of adsorbed water, b) the mass loss of crystal water, c) the phase transition from $h$-$MoO_3$ to $\alpha$-$MoO_3$[25]. However, it shows the ignorable mass loss in $\alpha$-$MoO_3$ with increasing temperature, suggesting that there is no crystal water in $\alpha$-$MoO_3$ (Supplementary Fig. 5). According to the high-resolution X-ray photoelectron spectroscopy (XPS) results, the peaks at a binding energy of 236.3 and 233.1 eV are attributed to Mo $3d_{3/2}$ and Mo $3d_{5/2}$ of Mo$^{6+}$, respectively (Supplementary Fig. 6). Therefore, the molecular formulae of as-synthesized $h$-$MoO_3$ could be defined as $h$-$MoO_3 \cdot 0.7H_2O$. It is well known that structural water plays an important role in accelerating the solid-state transport rate of charge carriers and enhancing the charge storage ability of electrodes, especially for non-metallic charge carriers such as H$^+$ and $NH_4^+$ [24,43].

### Electrochemical energy storage analysis

To investigate proton storage performance, the electrochemical performance of $MoO_3$ electrodes in the three-electrode system is evaluated in 0.5 M $H_2SO_4$. Cyclic voltammetry (CV) curves of the $h$-$MoO_3$ electrode exhibit a rectangular-like shape with a broad redox peak in the potential window of −0.53 to 0 V, which is regarded as a typical feature of pseudocapacitive behavior (Fig. 2a). The CV curves of $h$-$MoO_3$ are also measured in $H_2SO_4$ with various pH values at a scan rate of 2 mV s$^{-1}$ (pH from ca. 0 to 3). As Supplementary Fig. 7 shows, the oxidation peak shifts to higher potential and the reduction peak shifts to lower

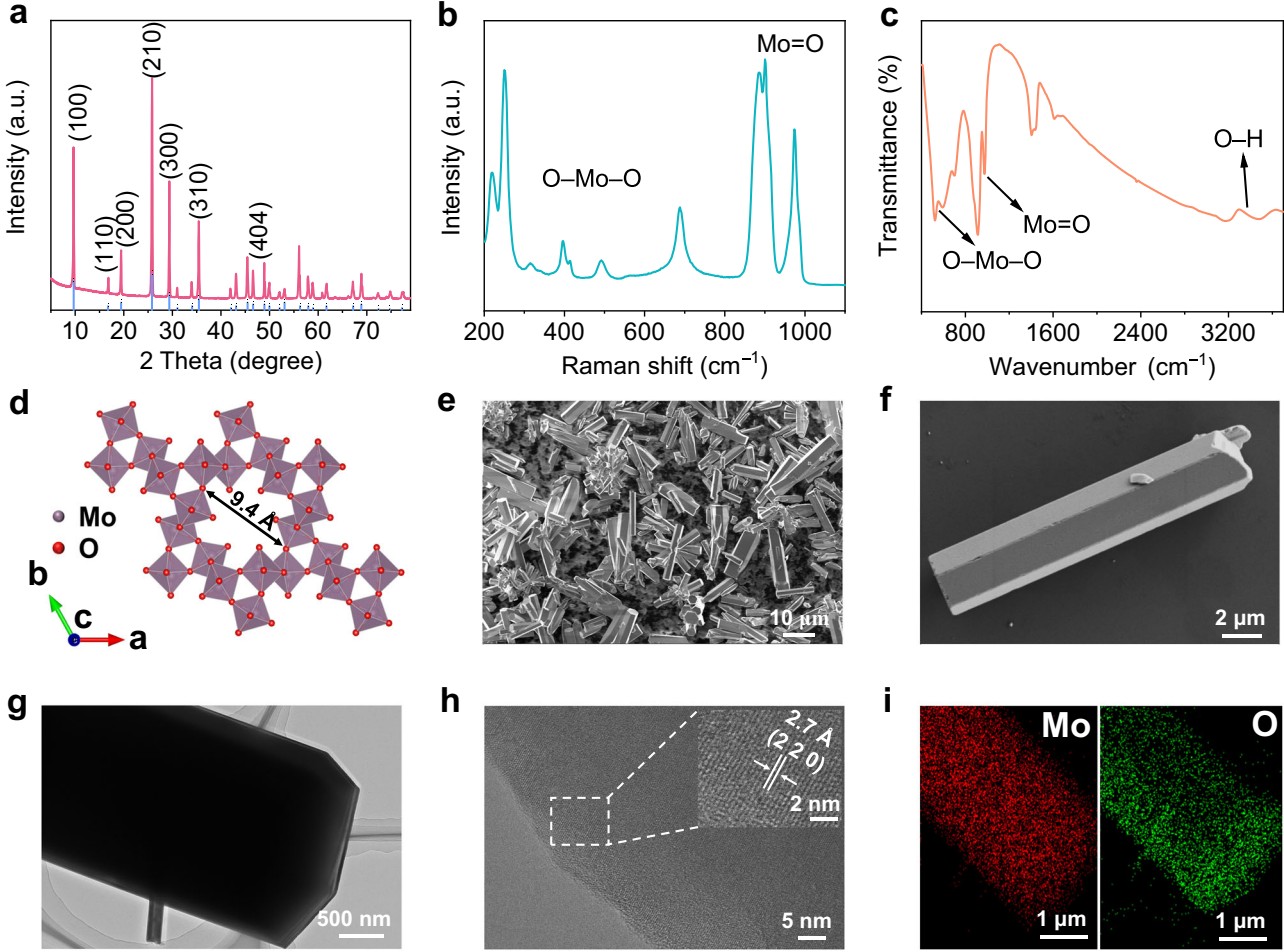

**Fig. 1 | Structure and morphology characterization of _h_-MoO₃.** **a** XRD pattern of _h_-MoO₃. **b** Raman spectra of _h_-MoO₃. **c** FTIR spectra of _h_-MoO₃. **d** Crystal structure of _h_-MoO₃. **e**, **f** SEM images of _h_-MoO₃ microrods. **g** TEM images of _h_-MoO₃ microrods. **h** HRTEM images of _h_-MoO₃ microrods (the inset is magnified lattice spacing). **i** Element mapping images of Mo and O.

potential, demonstrating that the redox potential of _h_-MoO₃ is pH-dependent. When the pH value of the H₂SO₄ solution is ≥ 3, an interesting feature shows negligible current response, which is attributed to the fact that the decrease of active materials involved in the electrochemical reaction with the reduced number of protons causes a decrease in the specific capacitance. At 2 mV s⁻¹, a series of protons can be reversibly intercalated into _h_-MoO₃ for a specific capacitance of 569 F g⁻¹ in 0.5 M H₂SO₄. When the scan rate is increased by a factor of 50 to 100 mV s⁻¹, the _h_-MoO₃ electrode still maintains a specific capacitance of 235 F g⁻¹. Owing to the size effect and intrinsic conductivity of electrode materials, the capacitance retention of _h_-MoO₃ electrodes at a high rate is not as high as well-designed nanostructured and carbon-modified pseudocapacitive materials. Nevertheless, the micrometer-sized _h_-MoO₃ electrodes achieve an acceptable capacitance at high rates in comparison with typical pseudocapacitive materials recently reported (Supplementary Table 1). As far as we know, there are few reported pseudocapacitive materials achieving excellent electrochemical performance in such large particle sizes. Additionally, the triangle-shaped galvanostatic charge and discharge (GCD) curves of _h_-MoO₃ electrodes also correspond to CV curves and show a similar shape even at a large current density, demonstrating a fast kinetics process for _h_-MoO₃ electrodes (Fig. 2b). In contrast, two pairs of sharp redox peaks in CV curves of _α_-MoO₃ indicate a typical battery reaction behavior (Supplementary Fig. 8a). The proton storage reaction for _α_-MoO₃ electrodes occurs in two pairs of charge and discharge plateaus from 0.3 V to −0.5 V, which is associated with different phase transition

stages (Supplementary Fig. 8b)[44-46]. Generally, the micrometer-sized _h_-MoO₃ not only decreases side reactions because of the small contact area between microparticles and electrolytes, but also increases the packing density of electrodes and saves the physical space of electrochemical energy storage devices in practical applications. As a result, a state-of-the-art volumetric capacitance of -1750 F cm⁻³ is obtained at 2 mV s⁻¹ (Fig. 2c), surpassing most electrode materials previously reported (Fig. 2d), such as two-dimensional transition metal carbides/nitrides (MXenes)[47-52], metal sulfides[53-56], metal oxides[57-66], metal nitrides[67-69], organic frameworks[70-72], redox Graphene (rGO)[73-75], conductive polymers[76-78], and activated carbon[79-83], and their detailed electrochemical performance is shown in Supplementary Table 2. It is important to note that while the volumetric capacitance of these reported nanomaterials can be enhanced by various structural designs, there are still several disadvantages associated with using nanomaterials, including high cost, complexity, and increased side effects. The bulk phase _h_-MoO₃ solves most drawbacks of nanometer-sized electrode materials and realizes unprecedented intrinsic pseudocapacitance at micrometer scales. The extraordinary electrochemical performance of _h_-MoO₃ originates from one-dimensional ionic and electronic pathways in the crystal structure, which enable protons and electrons to be transported quickly and steadily along the oriented highway, leading to _h_-MoO₃ electrodes with excellent capacitance and rate.

To evaluate the structural stability of _h_-MoO₃ and _α_-MoO₃ electrodes, thousands of cycles of two electrodes are measured at a current density of 20 A g⁻¹. _α_-MoO₃ electrodes show almost no capacity

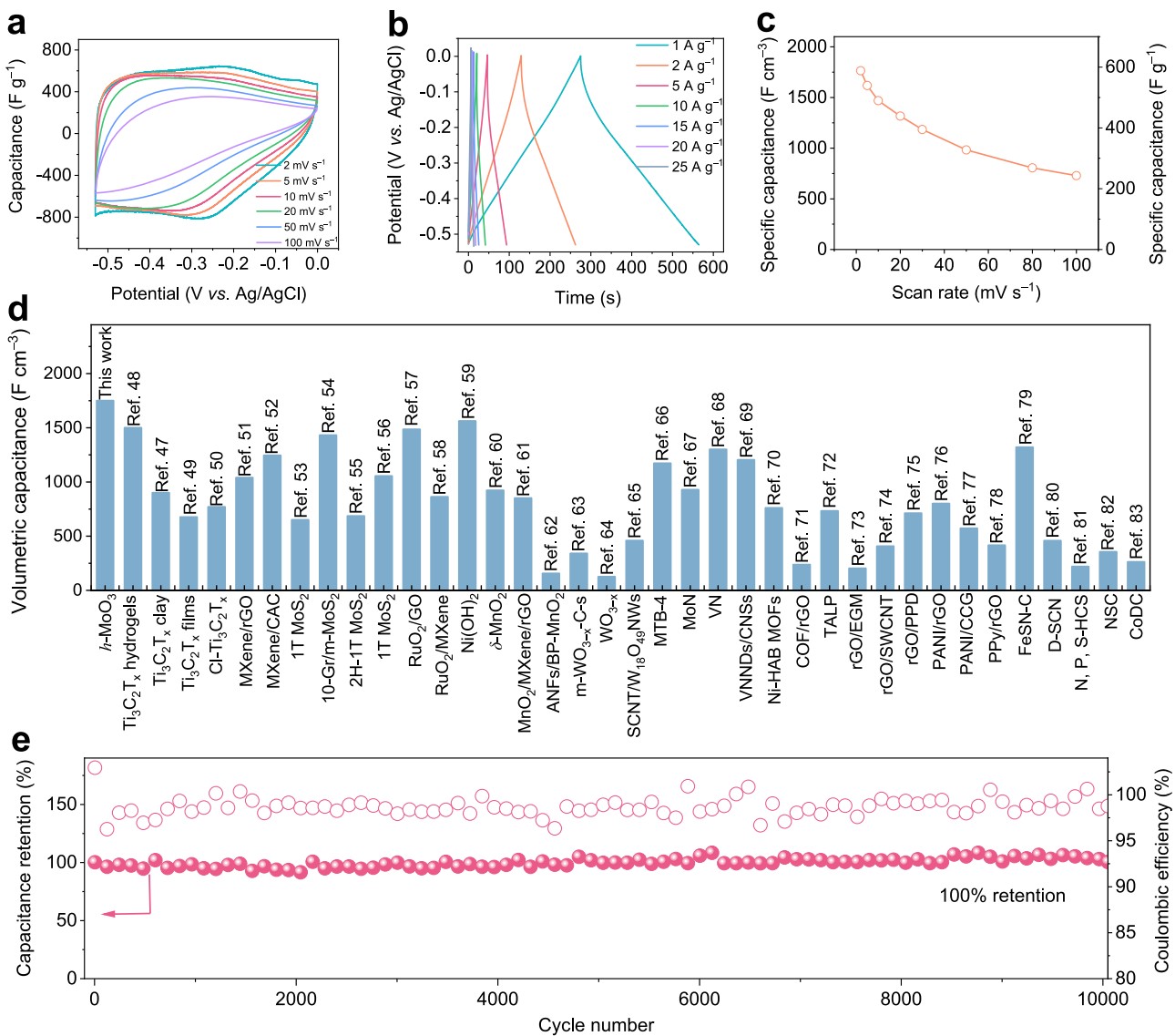

**Fig. 2 | Electrochemical performance of *h*-MoO₃ electrodes. a** Capacitance of *h*-MoO₃ electrodes at various scan rates. **b** GCD curves of *h*-MoO₃ electrodes at different current densities. **c** Volumetric capacitance of *h*-MoO₃ electrodes at different scan rates. **d** Maximum volumetric capacitance for several electrode materials. **e** Cycling stability performance for *h*-MoO₃ electrodes at 20 A g⁻¹.

after only 1000 cycles, which could be ascribed to irreversible capacity loss derived from structural distortion during charging and discharging (Supplementary Fig. 9). Remarkably, owing to the existence of suitable ion channels and ignorable structural strain during the electrochemical reaction, *h*-MoO₃ electrodes retain a capacitance retention rate of 100% over 10,000 cycles at 20 A g⁻¹ (Fig. 2e). Compared with the layered structure of *α*-MoO₃, these tunnel structures in *h*-MoO₃ generally exhibit extraordinary open superstructure to avoid the crystal structure transitions. The better durability of *h*-MoO₃ electrodes is also demonstrated by inductively coupled plasma optical emission spectrometry (ICP-OES) results in Supplementary Table 3, where the Mo content in the electrolyte for *h*-MoO₃ and *α*-MoO₃ after 1000 cycles is 69.9 and 123 mg L⁻¹, respectively. To clarify the charge storage mechanism of *h*-MoO₃, the power-law relationship between peak current and scan rate was explored:

$$i = av^b \tag{1}$$

where a and b are variable parameters. The b-value of 1 is considered a surface-controlled process, while the b value of 0.5 indicates a diffusion-controlled process[16]. In the scan range of 1 mV s⁻¹ to 100 mV s⁻¹, the cathodic and anodic b value are 0.8 and 0.87, respectively, demonstrating a surface-controlled pseudocapacitive mechanism for proton storage in *h*-MoO₃ electrodes (Supplementary Fig. 10a and b). The capacitive capacity contribution and diffusion capacity contribution at various scan rates are also calculated, in which the capacitive capacitance contribution exceeds 70% at a scan rate of 5 mV s⁻¹ and the diffusion capacity contribution gradually decreases with the increase of scan rates (Supplementary Fig. 10c and d). Nanostructured *h*-MoO₃ with a particle size of about 500 nm to 1 μm are prepared to the reveal size effect and charge storage characteristic (Supplementary Fig. 11). Due to more active area and shorter ion transport lengths, nanostructured *h*-MoO₃ shows good specific capacitance (645 F g⁻¹ at 2 mV s⁻¹) and rate performance (306 F g⁻¹ at 100 mV s⁻¹) (Supplementary Fig. 12). However, the large specific surface of nanomaterials also provides more active sites for other side reactions (such as hydrogen evolution reaction), leading to poor Coulombic efficiency and cyclic stability (Supplementary Fig. 13). Subsequently, cyclic voltammetry (CV) curves for *h*-MoO₃ electrodes are carried out in electrolytes with

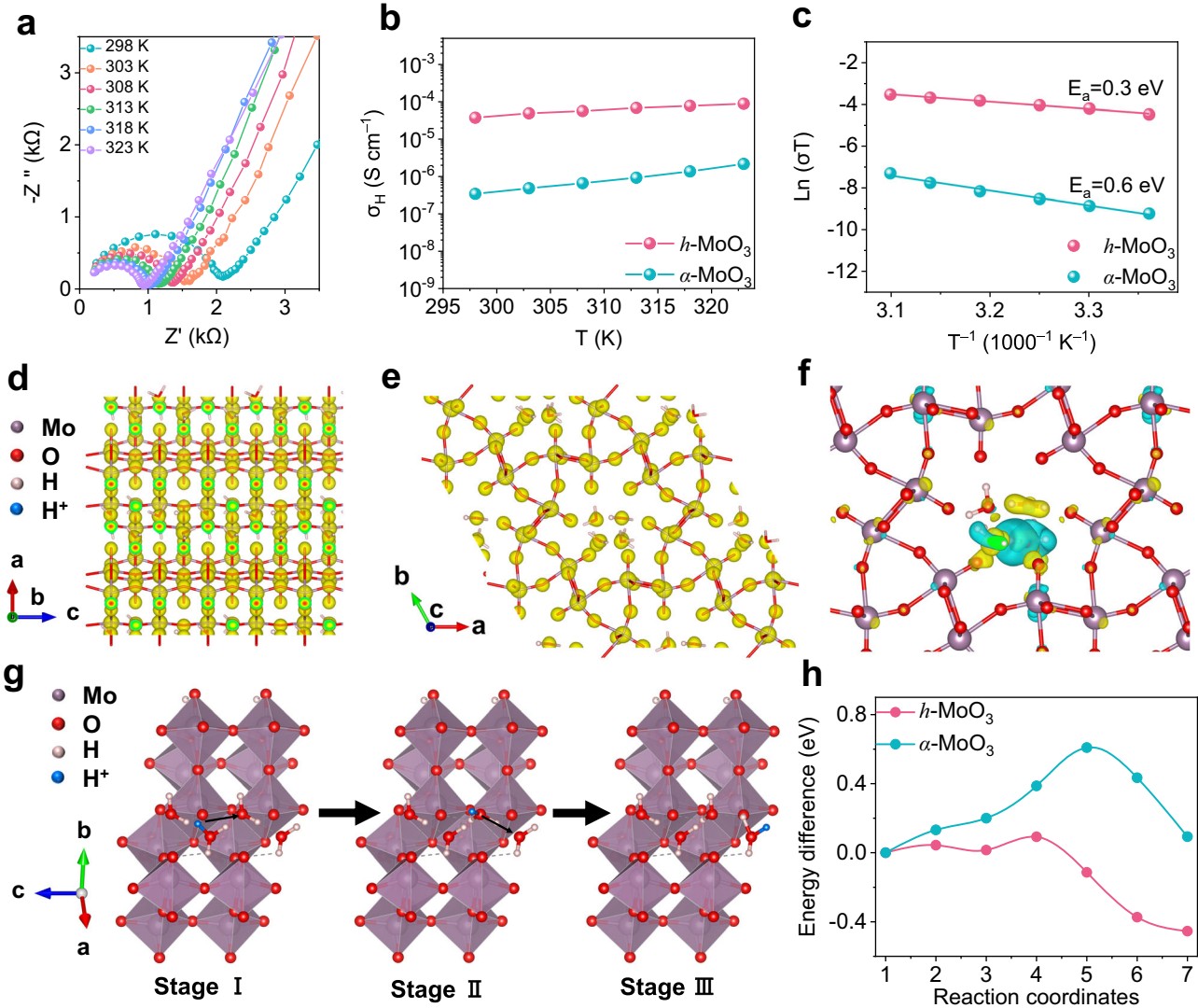

**Fig. 3 | Proton transport kinetics. a** Bulk impedance spectrum of $h$-MoO$_3$.
**b** Proton conductivity of $h$-MoO$_3$ and $\alpha$-MoO$_3$ at different temperatures.
**c** Activation energy of $h$-MoO$_3$ and $\alpha$-MoO$_3$. **d**, **e** Electron density distributions of $h$-MoO$_3$. **f** Charge density difference distribution after protons insertion into $h$-MoO$_3$.
**g** Migration pathway of protons in $h$-MoO$_3$. **h** Diffusion energy barrier of protons from stage II to stage III in $h$-MoO$_3$ and $\alpha$-MoO$_3$.

different charge carriers (such as Li$_2$SO$_4$, Na$_2$SO$_4$, K$_2$SO$_4$, and (NH$_4$)$_2$SO$_4$). All CV curves show negligible current response (Supplementary Fig. 14), suggesting that ion channels in $h$-MoO$_3$ are selective for H$^+$ instead of Li$^+$, Na$^+$, K$^+$, and NH$_4^+$. The characteristic of tiny electrical double-layer capacitance shows the absence of significant pseudocapacitance, which may be associated with the low active surface in micrometer-sized $h$-MoO$_3$. On the one hand, protons have the smallest ionic radius and mass that are favorable for fast transport in these channels, while Li$^+$, Na$^+$, K$^+$, and NH$_4^+$ with larger ionic radius and mass are more difficult to intercalate into electrodes. On the other hand, water molecules in ion channels can build a continuous hydrogen bond network for the ultra-fast synergistic transport of protons and water molecules (Li$^+$, Na$^+$, and K$^+$ might only be transported by slow ionic diffusion). Several $h$-MoO$_3$ electrodes with the active mass of 3, 5, and 10 mg cm$^{-2}$ are investigated for the practical application of electrode materials. The volumetric capacitance slightly decreases with increased active mass, but remains at 813 F cm$^{-3}$ even under 10 mg cm$^{-2}$ (Supplementary Fig. 15a). The rate-dependent fading tendency also occurs with the increase of the $h$-MoO$_3$ mass loading. For pursuing ultrahigh areal capacitance towards miniaturized devices, the areal capacitance of $h$-MoO$_3$ electrodes at

1 mV s$^{-1}$ is 1245, 1707, and 2622 mF cm$^{-2}$, respectively (Supplementary Fig. 15b). When increasing the scan to 20 mV s$^{-1}$, the area capacitance of over 600 mF cm$^{-2}$ is achieved in high-mass loading. Bulk $h$-MoO$_3$ electrode materials show potential value for designing thick electrodes with exceptional electrochemical performance and constructing supercapacitors with high volumetric and areal energy densities.

**Proton transport kinetics**

The electrochemical impedance spectrum (EIS) for $h$-MoO$_3$ with a temperature range of 298 K to 323 K is shown in Fig. 3a and displays an obvious temperature-dependent relationship. The semicircle in the high-frequency region represents the bulk impedance which indicates the transport rate of protons in the $h$-MoO$_3$ bulk phase (Supplementary Fig. 16). At different temperatures, the bulk impedance of $h$-MoO$_3$ is significantly smaller than that of $\alpha$-MoO$_3$, demonstrating that protons migrate more quickly in the $h$-MoO$_3$ bulk phase (Supplementary Fig. 17). Calculated proton conductivity of $h$-MoO$_3$ and $\alpha$-MoO$_3$ are $3.75 \times 10^{-5}$ and $3.43 \times 10^{-7}$ S cm$^{-1}$ at 25 °C, respectively. The proton conductivity increases with increasing temperatures, where the proton conductivity of $h$-MoO$_3$ is two orders of magnitude higher than that of $\alpha$-MoO$_3$ at different

temperatures (Fig. 3b). The reason is that the open tunnel structure for $h$-MoO$_3$ allows more protons to insert/de-insert quickly, while limited space for $\alpha$-MoO$_3$ can only accept a few protons slowly. According to the Arrhenius behavior related to proton conductivity at different temperatures, the activation energy of $h$-MoO$_3$ and $\alpha$-MoO$_3$ can be determined to assess the proton transport kinetics of these two materials (Fig. 3c). It is found that the activation energy barrier for the $h$-MoO$_3$ pellet does give a low value of 0.3 eV, in which protons move quickly through the Grotthuss conduction mechanism[33,34]. It is notable that a much higher activation energy for proton diffusion in the $\alpha$-MoO$_3$ is obtained, which is consistent with the slow Vehicle transport mechanism reported in some literature[84].

On the one hand, the $h$-MoO$_3$ features two different infinite one-dimensional ion pathways that are similar to high-rate Nb$_{16}$W$_5$O$_{55}$ along the c-axis[6], with the additional four-sided channels (Supplementary Fig. 18a, b). Similar to proton channels in proteins, structural water in the $h$-MoO$_3$ hexagonal channels can connect with each other to form a hydrogen bond chain along the c-axis direction, where protons may jump rapidly along the hydrogen bonds activating the Grotthuss mechanism without tunnel-blocking[34]. On the other hand, the layered structure of $\alpha$-MoO$_3$ is unable to provide charge carriers with sufficient space for fast transport and storage, causing the reaction kinetics of $\alpha$-MoO$_3$ to be limited by the bulk diffusion process of protons[20]. To demonstrate the above hypothesis, we evolve the migration process of protons in $h$-MoO$_3$ and $\alpha$-MoO$_3$ using theoretical calculations, in which MoO$_3$·0.7H$_2$O is selected as the standard model. Due to the strong electronic conductivity of the MoO$_6$ octahedra structure, electrons and protons can transfer from the redox sites to the conducting Mo-Mo chains (Fig. 3d). High electron conductivity is also maintained in the hexagonal tunnels, which may facilitate rapid proton transport along the water chains (Fig. 3e). One proton is first moved into the crystal structure of $h$-MoO$_3$, then the proton will unexpectedly migrate to water molecules to form H$_3$O instead of Mo$-$O$-$H after relaxation, suggesting that H$_2$O molecules are low-energy binding sites for protons (Supplementary Fig. 19). The charge density difference distributions of inserted protons are calculated to indicate that the bonding behavior is thermodynamically stable (Fig. 3f). Then, the proton will jump along the hydrogen bond to water molecule H$_2$O$^{II}$ (Stage II), and finally to H$_2$O$^{III}$ (Stage III), leading to a fast mass transport process (Fig. 3g). Nevertheless, the proton would initially form Mo$-$O$-$H in $\alpha$-MoO$_3$ (Stage I) and then slowly diffuse to MoO$^{II}$, finally to Mo$-$O$^{III}$ (Supplementary Fig. 20). The diffusion energy barriers for state II to III in $h$-MoO$_3$ and $\alpha$-MoO$_3$ are 0.09 and 0.61 eV, respectively, showing the more rapid ion transport kinetics of protons in $h$-MoO$_3$ (Fig. 3h).

## Structural evolution and energy storage mechanism

The in situ XRD measurements are performed to examine the changes of crystal structures of electrode materials during electrochemical reaction processes. An irreversible phase transition in the $h$-MoO$_3$ electrode is observed during the initial discharge, and the reaction process can be separated into three typical discharge stages (Fig. 4a). The changes in the position and intensity of corresponding XRD diffraction peaks are shown to evolve the lattice parameters and volume variation at different potential (Fig. 4b). For stage I, all diffraction peaks shift to lower angles with constant intensity, which corresponds to the expansion of lattice planes caused by protons intercalation. When it comes to stage II (the main stage of the irreversible phase transition reaction), the $h$-MoO$_3$ phase reduces in intensity with increasing protonation and peak shifts towards higher angles are observed corresponding to a shrinkage of the lattice plane. On still further proton intercalation (stage III), all diffraction peaks gradually shift to lower angles again, with no changes to the peak intensity. The three distinct mechanisms for changing the crystal structure suggest that the first inserted

protons might bond with different proton binding sites, leading to various adjustments in the lattice parameters. Figure 4c–e show the changes in the position and intensity of the diffraction peak (210), (300), (220), (310), (320), (410), and (404), which are completely consistent with the above reaction processes. During the first charge and the second discharge (Fig. 4f), the intensity of all diffraction peaks no longer changes, and no new peaks generate or disappear, indicating that the subsequent reaction was a reversible solid-solution in nature (Fig. 4g). It should be noted that the diffraction peak (210) only slightly shifts to a high angle with a decrease of the lattice parameter during the first cycle of protons de-insertion, and returns to the initial angle during the subsequent discharge process, demonstrating that the subsequent reaction is a highly reversible stable structural reaction based on proton intercalation/de-intercalation (Fig. 4h). Furthermore, the followed protons deintercalation/re-intercalation processes will only cause an ignorable structural strain of the crystal lattice (quasi-zero strain structural change), which may be ascribed to the fact that protons are transported rapidly along these fast one-dimensional ion channels without damaging the crystal structure of $h$-MoO$_3$. The structural evolution and degradation mechanism of $\alpha$-MoO$_3$ electrodes is evaluated using in situ XRD measurements. During the initial discharge, the new diffraction peaks at 50−55° appear, while some diffraction peaks (such as (110)) almost disappear, suggesting the formation of a new phase H$_x$MoO$_3$ (Supplementary Fig. 22a–c). During charging and the following discharging, a typical new/old phase transition reaction occurs, accompanying the intercalation/deintercalation of protons in $\alpha$-MoO$_3$ (Supplementary Fig. 22d, e). Particularly, the obvious diffraction peak shift during the electrochemical reaction is associated with the strong electrostatic interaction between charge carriers and electrode hosts, resulting in huge internal strain and the structure collapse of electrodes.

Based on the above proton conductivity and in situ XRD tests, a series of theoretical calculations are applied to reveal the proton transport pathway and storage sites of $h$-MoO$_3$ electrodes. With the $h$-MoO$_3$·0.7H$_2$O crystal structure being chosen as the calculation model, protons are intercalated into the electrode bulk phase and mainly exist in three compositions after the first discharge process. In stage I, protons will combine with crystal water to form hydronium−H$_3$O. In stage II, H bonds further with H to form H−H and fills in hexagonal tunnels. In stage III, oxygen atoms of the MoO$_6$ octahedra can be occupied, accompanied by H bonding to O as O−H to form H$_x$MoO$_3$ (Fig. 5a). Based on theoretical calculations, it reveals the proton coordination probability of H$_x$MoO$_3$ (when x = 0.5, 1, 2) at the three active sites (Supplementary Fig. 21a, b and Fig. 5b), which corresponds to the above reaction process. The migration and coordination processes of different protons in $h$-MoO$_3$·0.7H$_2$O are shown in Supplementary Movie 1. During the first discharge process, 2 mol protons and 2 mol electrons are intercalated into 1 mol $h$-MoO$_3$·0.7H$_2$O to generate $h$-H$_2$MoO$_3$·0.7H$_2$O with a tungsten bronze phase structure, which is similar to the first discharge mechanism of $\alpha$-MoO$_3$[20,46]. The binding energy of formed H$_2$ and H$_3$O is too high to kick out free protons in the subsequent charging process (Fig. 5c), leading to 1.34 mol hydrogen ions permanently occupying the internal space of $h$-H$_{1.34}$MoO$_3$·0.7H$_2$O. $h$-H$_2$MoO$_3$·0.7H$_2$O undergoes a structural self-optimization process to generate 0.66 mol protons acted as charge carriers. Recent studies have shown that structural self-improved transformation is found in some vanadium oxides and salt-rock phase materials, which can significantly optimize the electronic structure of the materials and further enhance the electrochemical performance of the electrode materials[85–87]. Then after the electrode is charged to 0 V, the protons binding with H$_x$MoO$_3$ (about 0.66 mol protons) can be de-intercalated from $h$-H$_2$MoO$_3$·0.7H$_2$O to form $h$-H$_{1.34}$MoO$_3$·0.7H$_2$O, and are reversibly inserted/de-inserted during following charging/discharging reaction to achieve outstanding structural stability. The first discharge

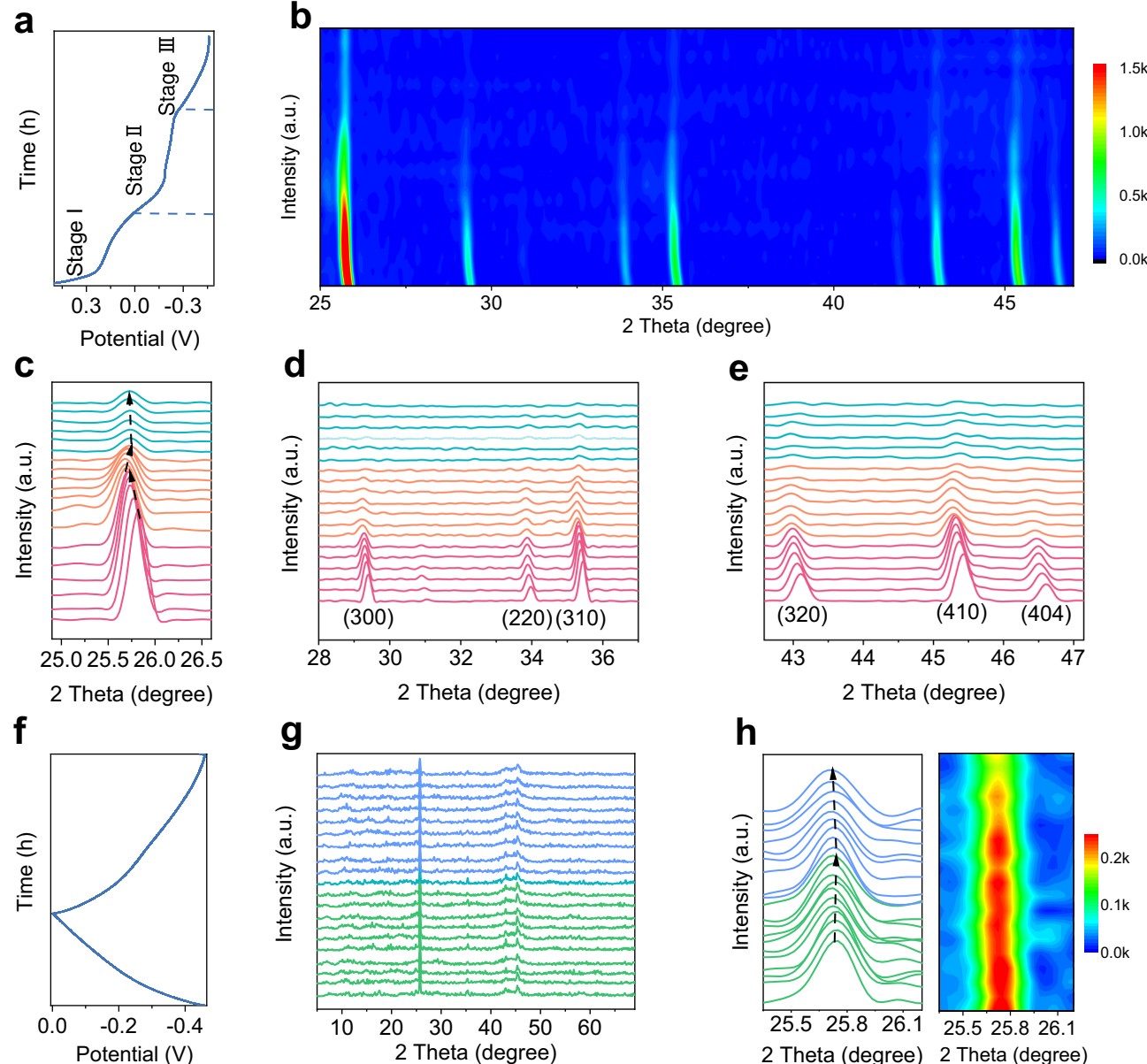

**Fig. 4 | Structural evolution of $h$-MoO$_3$. a** The first discharge curve of $h$-MoO$_3$ (a current density of 0.02 A g$^{-1}$). **b** Contour plots of in situ XRD during the first discharge process. **c–e** In situ XRD patterns of different diffraction peaks during the first discharge. **f** GCD curves of $h$-MoO$_3$ (the first charge and second discharge processes). **g** In situ XRD patterns during charge and discharge processes. **h** In situ XRD patterns of (210) diffraction peak during charge and discharge processes.

process is as follows:

$$MoO_3 \cdot 0.7H_2O + 0.63H^+ + 0.63e^- \rightarrow H_{0.63}MoO_3 \cdot 0.7H_2O \quad (2)$$

$$H_{0.63}MoO_3 \cdot 0.7H_2O + 0.71H^+ + 0.71e^- \rightarrow H_{1.34}MoO_3 \cdot 0.7H_2O \quad (3)$$

$$H_{1.34}MoO_3 \cdot 0.7H_2O + 0.66H^+ + 0.66e^- \rightarrow H_2MoO_3 \cdot 0.7H_2O \quad (4)$$

In the subsequent charging and discharging process, the reaction mechanism is described according to the following equation:

$$H_2MoO_3 \cdot 0.7H_2O \leftrightarrow H_{1.34}MoO_3 \cdot 0.7H_2O + 0.66H^+ + 0.66e^- \quad (5)$$

The size-independent intrinsic pseudocapacitance of bulk $h$-MoO$_3$ originates from the proton intercalation pseudocapacitance

mechanism, differing from nanostructured RuO$_2$ based on surface redox pseudocapacitance mechanism[28,32]. Furthermore, crystal structure and confined fluids are uncovered to play a critical role in enabling the remarkable transport and storage of non-metallic ions.

**Full energy storage device**
The Cu$_{0.82}$Co$_{0.18}$HCF electrode we previously reported was used as the cathode to construct an asymmetric proton pseudocapacitor device for the practical application of $h$-MoO$_3$ electrodes (Fig. 6a)[88]. The potential window of anode (−0.53 to 0 V) and cathode (0.2 to1.1 V) broaden the working voltage window of this device up to 1.6 V. When the scan rate is increased from 2 to 50 mV s$^{-1}$, the symmetrical CV curves imply the excellent rate performance of the proton pseudocapacitor (Fig. 6b). Particularly, this proton pseudocapacitor delivers a maximum specific capacitance of 95 F g$^{-1}$ (based on the total mass of anode and cathode materials) at a scan rate of 2 mV s$^{-1}$ because of complementary anode and cathode materials with outstanding kinetic

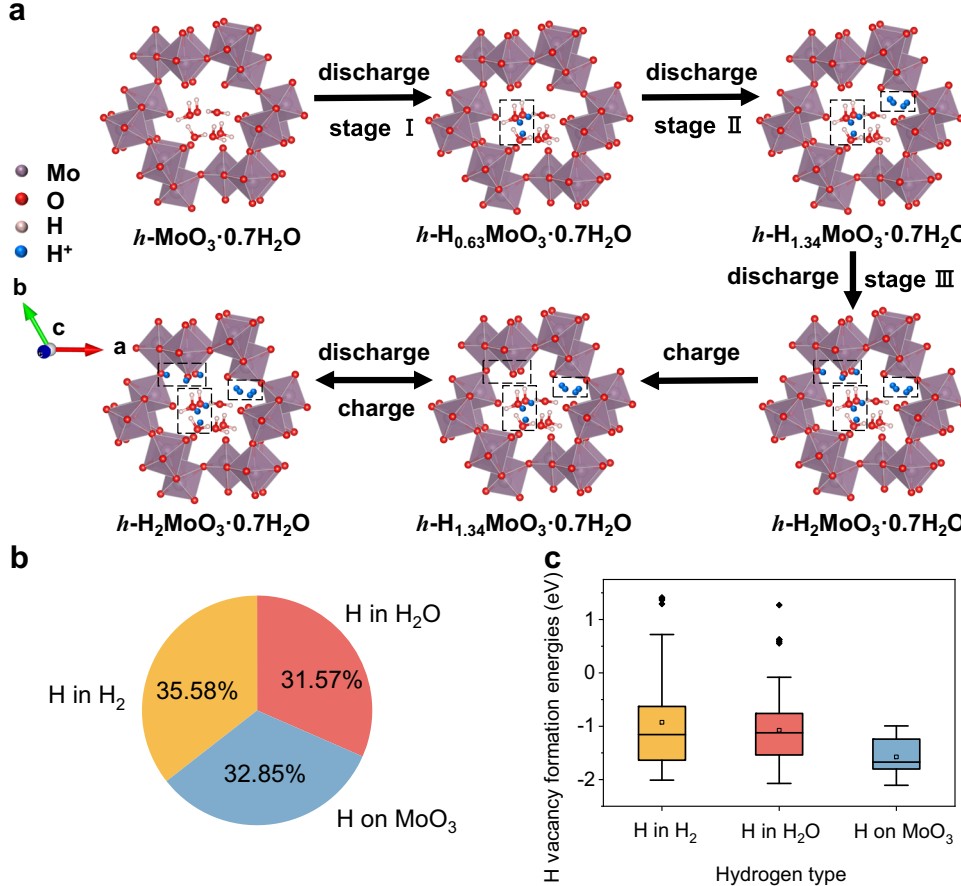

**Fig. 5 | Energy storage mechanism of *h*-MoO₃. a** Schematic depiction of the energy storage mechanism in *h*-MoO₃·0.7H₂O. **b** The distribution of H coordination environments from the room temperature AIMD simulations of *h*-H₂MoO₃·0.7H₂O structure. **c** The formation energies of H vacancy with three different H coordination environments of *h*-H₂MoO₃·0.7H₂O structure.

properties (Fig. 6c). At 50 mV s⁻¹, the device is still able to deliver a specific capacitance of about 71 F g⁻¹ (75% of the initial specific capacitance). Owing to the pseudocapacitive nature of electrode materials, the "mirror-like" GCD curves at different current densities indicate an ideal kinetics behavior (Fig. 6d)[89]. The asymmetric device exhibits a maximum energy density of 36 Wh kg⁻¹ at a power density of 1.0 kW kg⁻¹, and remains a high energy density of 21 Wh kg⁻¹ at a maximum power density of 21 kW kg⁻¹, which will maximize the energy and power density of supercapacitors (Fig. 6e). The electrochemical performance of this proton pseudocapacitor (especially the energy density at maximum power density) surpasses not only most supercapacitors in acid electrolytes, but also many typical supercapacitors in alkaline and neutral electrolytes (Supplementary Table 4), such as 2-GCE//AC (23.4 Wh kg⁻¹ at 3.9 kW kg⁻¹ and 21.1 kW kg⁻¹ at 10 Wh kg⁻¹) and 1@Ti₃C₂Tₓ//ACL (32.2 Wh kg⁻¹ at 2.4 kW kg⁻¹ and 12 kW kg⁻¹ at 19.7 Wh kg⁻¹) in acid electrolytes[90,91], MLMO//MLMO (34.1 Wh kg⁻¹ at 0.8 kW kg⁻¹ and 14.5 kW kg⁻¹ at 19.4 Wh kg⁻¹) in alkaline electrolytes[92], and MoO₃@CNT//MnO₂@CNT (27.8 Wh kg⁻¹ at 0.52 kW kg⁻¹ and 10 kW kg⁻¹ at 9.8 Wh kg⁻¹) in neutral electrolytes[93]. Besides, the proton pseudocapacitor exhibits outstanding cyclic stability for long-term application in electronic devices, delivering 83% of its initial specific capacitance after 10,000 cycles at a current density of 10 A g⁻¹ (Fig. 6f).

## Discussion

In conclusion, a bulk *h*-MoO₃ electrode material with selective proton channels has been designed without nanoscaling, to provide stable host structures for proton intercalation and storage, thus achieving a

high specific capacitance of 569 F g⁻¹ (no capacitance loss after 10,000 cycles). Micrometer-sized superstructure motifs of *h*-MoO₃ effectively enhance the packing density of electrodes and avoid the negative effects of nanostructures, resulting in a state-of-the-art volumetric capacitance (-1750 F cm⁻³). Fast ion channels enhance the proton conductivity of *h*-MoO₃ by two orders of magnitude compared with that of α-MoO₃, which significantly improves the migration number and rate of protons. Protonation-induced structural rearrangement optimizes the crystal structure of *h*-MoO₃ electrodes and realizes rapid reversible solid-state proton transport and storage. Besides, the proton pseudocapacitor based on *h*-MoO₃ achieves an exceptional energy density at an ultra-high power density. This work offers novel insights into electrode structure design principles for bulk materials with fast chargeability and structural stability, and brings light to the transport and storage mechanism of non-metallic charge carriers.

## Methods

### Synthesis of *h*-MoO₃

*h*-MoO₃ was prepared by a simple hydrothermal method[25,94]. Typically, 1 g Mo powders were added into 30 mL hydrogen peroxide (30 wt%) solution with rapid stirring in an ice-water bath. After continuous stirring for 1 hour, 35 mL water and 0.56 g NH₄NO₃ were added to the solution. The mixture was transferred to a 100 mL Teflon-lined autoclave (Anhui Kemi Machinery Technology Co., Ltd) and heated at 150 °C for 12 hours. After cooling to room temperature, the samples were centrifuged and washed with water and ethanol several times, and then dried in a vacuum drying at 60 °C overnight. Nanostructured *h*-MoO₃ were prepared by mechanical ball milling using a planetary ball mill at

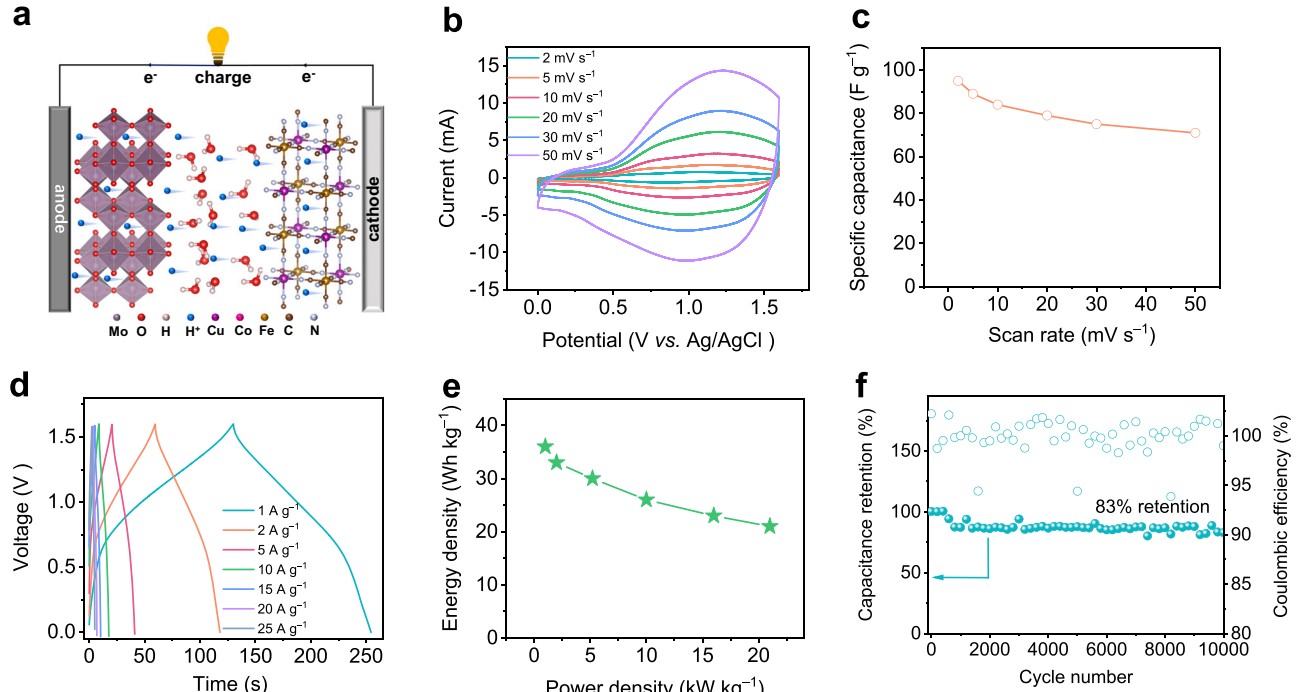

**Fig. 6 | Electrochemical performance of proton pseudocapacitors. a** Schematic diagram of the full proton pseudocapacitor device. **b** CV curves at various scan rates. **c** Specific capacitance at different scan rates. **d** GCD curves at different current densities. **e** Comparison of energy and power density. **f** Capacitance retention at 10 A g⁻¹.

450 rpm for 20 h in an argon atmosphere. For the preparation of $\alpha$-MoO₃, all experimental procedures and parameters were the same except that 0.56 g NH₄NO₃ was replaced by 0.5 mL HNO₃ (65 wt%).

**Physical characterization**

XRD patterns of samples were obtained using a Bruker D8-A25 diffractometer with Cu Kα radiation (λ = 1.5406 Å). Raman spectra were recorded on a Raman Spectrometer (LabRAM HR Evolution) with a 532 nm laser. FTIR spectra were measured on a Bruker INVENIO spectrometer with an ATR unit. SEM (TESCAN LYRA3), TEM (Tecnai 12), and XPS (KRATOS AXIS SUPRA) were used to analyze the microscopic morphology and chemical composition of materials. TGA measurement was performed by a Mettler Toledo TGA/490PC thermobalance from 25 °C to 700 °C in steps of 10 °C under an N₂ flow. ICP-OES (Thermo ICAP PRO) was used to analyze Mo elements in electrolytes.

**Electrochemical measurements**

The electrode was consisted of 70 wt% active material, 20 wt% acetylene black, and 10 wt% polyvinylidene fluoride (PVDF). The components were mixed in 1-methyl-2-pyrrolidinone (NMP) to obtain a slurry in a mortar, then this slurry was coated on graphite paper with a diameter of 1.2 cm (the active mass loading of ~1 mg cm⁻²) and dried at 60 °C overnight. In particular, active material, acetylene black, and polytetrafluoroethylene at a ratio of 7:2:1 was rolled to manufacture high-loading free-standing electrodes. All electrochemical performances for electrodes and devices were evaluated at 25 °C in the same laboratory. For three-electrode measurements, all electrochemical measurements were conducted using plastic Swagelok cells (active materials, free-standing activated carbon films, and Ag/AgCl electrodes were used as working electrodes, counter electrodes, and reference electrodes). In addition, 0.5 M H₂SO₄ and Whatman filter papers were used as electrolytes and separators, respectively. The two-electrode asymmetric proton pseudocapacitor was assembled using Cu₀.₈₂Co₀.₁₈HCF electrodes and *h*-MoO₃ electrodes as cathodes and anodes. The electrochemical performance of electrodes and asymmetric supercapacitors was tested using cyclic voltammetry (CV) and galvanostatic charging/discharging (GCD) on a CS2350H electrochemical workstation. Cycle stability of materials and devices was measured using the CT2001A Land Battery Test System.

**Proton conductivity test**

*h*-MoO₃/$\alpha$-MoO₃ powders (~300 mg) were poured into the mold, forming a pellet with a thickness of 1 mm and a diameter of 13 mm under pressure. Then the pellet was assembled into a CR2032 coin cell, where some glass fiber filter chips wetted by deionized water were placed into the cell to maintain 100% humidity. The EIS tests were recorded on a Biologic VMP-300 multichannel electrochemical workstation from 7 MHz to 100 mHz with a voltage amplitude of 20 mV, in which the test temperature was adjusted from 25 °C to 55 °C. Proton conductivity was calculated according to the following equation:

$$\sigma_H = \frac{t}{RS} \qquad (6)$$

Where $\sigma_H$, $R$, $t$, and $S$ represent the proton conductivity (S cm⁻¹), bulk resistance (Ω), thickness of pellet (cm), and contact area of pellet (cm²), respectively. According to proton conductivity under different temperature, the activation energy was calculated by:

$$\ln(\sigma_H T) = \ln A - \frac{E_a}{k_B T} \qquad (7)$$

Where $T$, $E_a$, $k_B$, and $A$ represent the temperature (K), activation energy (eV), Boltzmann's constant (1.38 × 10⁻²³ J K⁻¹), and pre-exponential factor.

**In situ X-ray diffraction measurements**

The working electrode for in situ XRD testing consisted of active material, acetylene black, and polyvinylidene fluoride (PVDF) with a

weight ratio of 8:1:1 coated on carbon clothes. The electrode was placed in a modified three-electrode electrochemical cell, where free-standing activated carbon, Ag/AgCl electrode, and 0.5 M $H_2SO_4$ as the counter electrode, reference electrode, and electrolyte. Then the Kapton film was used as a test window to fasten the working electrode and transmit X-rays. The current density and potential window of the working electrode were controlled using the CT2001A Land Battery Test System and the changes of crystal structure were recorded in the X-ray diffractometer.

## Calculations

Gravimetric capacitance was obtained by integrating the discharge part of the CV curve according to the following formula:

$$C_g = \frac{\int i dV}{m v \Delta V} \tag{8}$$

Where $C_g$, $i$, $V$, $m$, $v$, and $\triangle V$ represent the gravimetric capacitance (F g$^{-1}$), current (mA), potential (V), the mass of active material (mg), scan rate (mV s$^{-1}$), and potential window (V), respectively. Notably, m is the total mass of the cathode and anode active materials for the two-electrode configuration. The packing density ($\rho$) was measured by determining the geometrical parameters and total electrode mass after h-$MoO_3$ powder without the addition of conductive carbon was coated on the current collector. Then volumetric capacitance ($C_V$) was calculated by using the following equation:

$$C_V = \rho C_g \tag{9}$$

To construct the proton pseudocapacitor, the mass of the cathode and anode was calculated according to the following formula:

$$\frac{m_{g_+}}{m_{g_-}} = \frac{C_- \times E_-}{C_+ \times E_+} \tag{10}$$

Where $m_{g_+}$, $C_+$, and $E_+$ represent the mass (mg), gravimetric capacitance (F g$^{-1}$), and potential window (V) of cathode. $m_{g_-}$, $C_-$, and $E_-$ represent the mass (mg), gravimetric capacitance (F g$^{-1}$), and potential window (V) of anode. Gravimetric energy density and gravimetric power density of asymmetric device were calculated according to the following formula:

$$E_g = \frac{\int IV dt}{3.6M} \tag{11}$$

$$P_g = \frac{3600E}{\Delta t} \tag{12}$$

where $E_g$, $P_g$, $M$, and $\Delta t$ represent the energy density (Wh kg$^{-1}$), the power density (kW kg$^{-1}$), the total mass of the cathode and anode (mg), and the discharge time (s).

## Computational parameters and models

All calculations were carried out by using the projector augmented wave method in the framework of the density functional theory (DFT)[95], as implemented in the Vienna ab-initio Simulation Package (VASP 5.4.4). The generalized gradient approximation (GGA) functional, Perdew-Burke-Ernzerhof (PBE) exchange-correlation functional, and projector-augmented wave (PAW) were used. The plane-wave energy cutoff was set to 450 eV, and the Monkhorst-Pack method was employed for the Brillouin zone sampling[96]. The convergence criteria of energy and force calculations were set to 10$^{-5}$ eV/atom and 0.01 eV/Å, respectively. The $k$-point mesh was set to $2 \times 2 \times 7$ for the $MoO_3$ unit cell. The geometry optimizations and electronic structures were calculated by using the spin-polarized method[97]. The

$MoO_3 \cdot 0.7H_2O$ structure was built by inserting four water molecules into the Z-axis channel of $MoO_3$ unit cell, and further optimized by the DFT calculations. The extra 9 H, 18 H, and 36 H atoms were inserted into the $MoO_3 \cdot 0.7H_2O$ structure to construct the $H_{0.5}MoO_3 \cdot 0.7H_2O$, $HMoO_3 \cdot 0.7H_2O$, and $H_2MoO_3 \cdot 0.7H_2O$ structure, respectively, and they were optimized by the DFT calculations. To further get a reasonable configuration of H and $H_2O$ inserted $MoO_3$, $H_{0.5}MoO_3 \cdot 0.7H_2O$, $HMoO_3 \cdot 0.7H_2O$, and $H_2MoO_3 \cdot 0.7H_2O$ structure were equilibrated by the room temperature (300 K) Ab-initio molecular dynamics (AIMD) simulations. To maintain the computational cost at a reasonable level, a smaller plane wave energy cut-off of 300 eV and gamma-centered k-point grids were chosen for AIMD simulations. The time step was set to 1 fs, and the system was equilibrated for 10000 steps with a total time of 10 ps in a statistical ensemble with a fixed particle number, volume, and temperature (NVT). The formation energy of H vacancy was calculated by the formula of $E_f = E(H\ vacancy) - E(pristine) - 0.5E(H_2)$, where E(H vacancy), E(pristine) and E($H_2$) represent the DFT energy of the $H_2MoO_3 \cdot 0.7H_2O$ structure with a H vacancy, the pristine $H_2MoO_3 \cdot 0.7H_2O$ structure and gas $H_2$ molecule, respectively. To verify the accuracy of GGA calculation results, non-empirical strongly constrained and appropriately normed (SCAN) meta-generalized gradient approximation (meta-GGA) was used to optimize and verify the vacancy formation energy of $H_2MoO_3 \cdot 0.7H_2O$. The results obtained by meta-GGA are in agreement with the GGA results, demonstrating the applicability of GGA and meta-GGA. For $\alpha$-$MoO_3$, the orthorhombic cell with space group Pbnm was used to optimize both the lattice and ion positions. To calculate crystal structure and diffusion energy barrier, supercells containing $2 \times 1 \times 2$ conventional cells were used, with the Monkhorst–Pack scheme $k$-point grid set to $5 \times 3 \times 5$. Additionally, the energy diffusion pathways of ions and the corresponding energy barriers were calculated using the climbing image nudged elastic band (CI-NEB) method.

## Data availability

All relevant data that support the findings of this study are presented in the manuscript and supplementary information file. Source data are available from the corresponding author upon reasonable request.

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

## Acknowledgements

This work was supported by the National Natural Science Foundation of China (52072173—L.S.), Jiangsu Province Outstanding Youth Fund (BK20200016—L.S.), Leading Edge Technology of Jiangsu Province (BK20202008—X.Z.), the Fundamental Research Funds for the Central Universities (No. ILA22061, ILA22075—L.S.), and Postgraduate Research & Practice Innovation Program of Jiangsu Province (KYCX23_0367—T.X.). The authors would also like to acknowledge the Center for Microscopy and Analysis at Nanjing University of Aeronautics and Astronautics for the physical characterization analysis and Neware Technology Limited for the electrochemical data analysis.

## Author contributions

L.S. supervised the project. L.S. and T.X. conceived the concepts for the research project. T.X. performed the material synthesis, electrochemical tests, data analysis, and composed the manuscript. Z.X. and T.Y. carried out the DFT calculation and analysis. T.X., M.Z. and D.C. assisted with the in situ XRD characterization. All authors co-discussed the results and commented the manuscript.

## Competing interests

The authors declare no competing interests.
