## [Peer Review File · Nature Communications]

Discovery of fast and stable proton storage in bulk hexagonal molybdenum oxideREVIEWER COMMENTS

Reviewer #1 (Remarks to the Author):

This work focused on the interesting finding for the fast proton transport phenomenon of a bulk hexagonal molybdenum oxide (h-MoO₃) with useful ion channels for energy storage. The topic is generally interesting to the potential readers in this journal but a major revision with considering below comments is required for the possible publication in this journal.

1. Introduction. From the literature, the electroactive materials with their charge storage mechanisms involving the exchange of protons have to be reviewed and compared in order to clarify whether the finding for the fast proton transport phenomenon of a bulk hexagonal molybdenum oxide (h-MoO₃) is new or not. Accordingly, the authors cannot ignore the typical references discussing the materials with the similar characteristics for pseudocapacitors. For example, RuO₂ in various hydrated degrees have been demonstrated to be the typical materials for pseudocapacitors with proton exchange (e.g., see *J. Phys. Chem. B* 106 (2002) 3592 and *Chem. Mater.* 19 (2007) 2112). In addition, the mixed electron and proton conductivity of hydrous RuO₂ have been discussed (*Langmuir* 15 (1999) 774, *Angew. Chem. Int. Ed.* 42 (2003) 4092, and *J. Phys. Chem. B* 109 (2005) 7330). The authors need to give a balanced report on the important references in the literature and compare/comment their idea with these typical references in order to emphasize and clarify their originality and novelty.

2. From the decay in capacitance with the scan rate or GCD current density, the capacitance retention is poor in comparison with the typical pseudocapacitor materials (e.g., *Angew. Chem. Int. Ed.* 42 (2003) 4092, *Nano Lett.* 6 (2006) 2690, and *Electrochim. Acta* 54 (2009) 4574 for RuO₂; *J. Mater. Chem. A* 5 (2017) 10021 and *J. Electrochem. Soc.* 166 (2019) A1875 for MxMnO₂; *J. Power Sources* 196 (2011) 2387 for WO₃). The authors have to compare and comment this property with suitable references because of the supercapacitor application.

3. For the high-rate performance, the scan range of 1 - 100 mV/s is acceptable for the supercapacitor application but not typically high in comparison with those using the very high scan rates of 500-1000 mV/s or even 10000 mV/s. In addition, the anodic and cathodic b-values significantly deviate from the ideal value, 1.0. Again, the authors have to compare and comment this property with suitable references because of the micrometer structure of their materials or other reasons. In my opinion, the authors have to prepare and compare the charge storage characteristics of nanostructured h-MoO₃ with those of bulk h-MoO₃ in order to emphasize their unique micrometer structure.

4. The authors have to compare the CVs measured in the aqueous media with various pH values in order to confirm the charge storage mechanism of h-MoO₃ (e.g., pH from ca. 0 to 3). In other words, the redox potential of h-MoO₃ should be pH-dependent. In addition, water is a proton acceptor and donor, why the CV curves of h-MoO₃ electrodes measured in neutral electrolytes do not show the pseudocapacitive responses in a certain degree (e.g., half of the charges) if water molecules in ion channels can build a continuous hydrogen bond network for the ultra-fast synergistic transport of protons? RuO₂ and IrO₂ with the mixed electron and proton conductivity have been found to possess the redox pseudocapacitance in aqueous media with a very wide pH range (ca. pH from 0 to 14, e.g., see *J. Electrochem. Soc.* 139 (1992) 2158). The authors have to comment the charge storage characteristics of h-MoO₃ in comparison with RuO₂ and IrO₂.

5. What is the gravity-based specific capacitance of h-MoO₃? From the CVs of h-MoO₃, the gravity-based specific capacitance of h-MoO₃ is similar to that of WO₃. However, the density of WO₃ is ca. 7.16 g/cm³, indicating that the volumetric capacitance of WO₃ (e.g., see *J. Power Sources* 196 (2011) 2387 and *Electrochem. Commun.* 12 (2010) 1800) is higher than

that of h-MoO₃. The authors have to compare and comment this concern in Fig. 2d.

6. The degradation mechanism of alpha-MoO₃ needs to be confirmed by direct evidences via various material characterization tools.

7. Do the authors make sure that h-MoO₃ can proceed the redox reactions if Li⁺, Na⁺, or K⁺ can exchange with h-MoO₃? In my opinion, if the redox reactions of h-MoO₃ do occur with the exchange of Li⁺, Na⁺, or K⁺, the active sites on the superficial region should contribute the pseudocapacitance in the neutral media. How about the authors opinions? The authors should give their opinions in the revision or supporting materials.

8. What is the definition of bulk resistance? How to obtain the bulk resistance from the EIS data? For EIS data, in the high frequency region (e.g., > 2.5 kHz), mass transfer can be neglected and the charge-transfer process at the outermost surface of the electrode in contact with the electrolyte will be dominant. This behavior is typical for pseudocapacitor materials. Why did the semicircle in the high frequency region represent the bulk impedance which indicates the transport rate of protons in the h-MoO₃ bulk phase?

9. The authors have to provide the equivalent circuit for fitting the EIS data and clarify all the elements in the equivalent circuit (e.g., see J. Phys. Chem. B 109 (2005) 7330). Note that the frequency was swept from 7 MHz to 100 mHz in this work; why did the impedance obtained at the high-frequency end not approach the real part axis in Fig. 3a? For most Faradaic reactions, the impedance obtained at the high-frequency end will approach the real part axis at ca. 50kHz – 1MHz (e.g., see Supplementary Fig. 9).

10. The water content of h-MoO₃·0.7H₂O is very high (Mo:H₂O = 1:0.7). In addition, I did not see the suitable ratio between Mo and water from the models shown in Supplementary Fig. 10 and 11. In addition, from the model shown in Supplementary Fig. 12, I cannot find the role of hydrated water favoring the transport of proton. Accordingly, the models proposed here are too arbitrary to be acceptable.

11. The asymmetric device exhibits a maximum energy density of 36 Wh/kg at a power density of 1.0 kW/kg, and remains 21 Wh/kg at 21 kW/kg. The performance is good but not very outstanding since most Ni-Co oxyhydroxides/sulfides-based asymmetric/hybrid cells (e.g., see J. Power Sources 253 (2014) 205) can provide the similar or even better performance than the cell demonstrated in this work. The authors need to make correct comparisons and clarify their merits of their material and cell designs from the typical references.

Reviewer #2 (Remarks to the Author):

The authors insist that a bulk hexagonal molybdenum oxide (MoO₃) with ion channels enables the fast transport of protons due to its having hydrogen bonding topochemistry, as demonstrated by several orders of magnitude higher than those of orthorhombic MoO₃. Besides, it achieves an volumetric capacitance (~1750 F/cm³) and long cycle life (>10,000 cycles). They also tried to unveil the mechanism through in-situ XRD and density functional theory calculations.

The synthesized methods for h-MoO₃ with open channels appear to have been already reported by References 25 and 26 so that this work is considered to report the electrochemical performances of h-MoO₃ with open channels originally reported through the previous studies. This work is interesting in views of volumetric capacitance, rate-capability, and cycle life, but its electrode structure is simply based on h-MoO₃ (similar to the previous studies) whose open channels are commonly expected to enable fast transports of protons and other ions leading to high power density as well as cycle stability. Furthermore, the high tap density of the hexagonal microrods is expected from the morphology from h-MoO₃.

Of course, the finding of h-MoO₃ with adsorbed waters to enhance proton transports appears to be very interesting, although similar mechanisms (For example, Reference 23) have been already reported.

Additionally, the authors did not describe the mass loadings (both three-electrode and also full-cell configurations) for their electrochemical performances, which is important to judge the novelty of performances. It is notable that electrochemical performances vary depending on different mass loadings. Whether their electrode could be practically usable as a commercial-type, their performances should be really demonstrated at high mass loadings.

Besides, the generalized gradient approximation (GGA) based on Perdew-Burke-Ernzerhof (PBE) exchange functional is subjected to the underestimation of kinetic energies (over-performing). Especially, proton transfer using PBE exchange could be overestimated using PEB. Also, the authors did not specify which correlation functional is used in their DFT calculations. Another issue is relating to their energies. The authors need to clarify whether their energies include vibration energies and thermal corrections or not.

In conclusion, the authors reported interesting performances of h-MoO₃ with open channels and adsorbed waters. However, this work suffers from technical issues. At first, the authors need to report their performances at different mass loadings, including high mass loadings for practical applications. Secondly, DFT calculations appear to be based on very simple approaches not ensuring the accurate descriptions of the systems. In addition to these technical issues, the reported electrode materials appear to be not on the first reports.

Reviewer #3 (Remarks to the Author):

The manuscript reported a type of bulk electrode material with both excellent volumetric capacitance for proton storage and cycling stability as compared with previously reported nanostructured material. There are some concerns to be addressed before its possible publication:

1. h-MoO₃ have been reported for pseudocapacitive energy storage with proton or NH₄⁺ as charge carrier through Grotthuss or Vehicle mechanism (Nanoscale 2015, 7, 11777; Adv. Mater. 2020, 32, 190780). The preparation method of this work is also based on Adv. Mater. 2020, 32, 190780.

2. As for the high volumetric performance, the calculation details of volumetric capacitance should be given. Why the author choose the potential window of -0.5 V to 0 V in three-electrode test? Compared to many literatures, the applied potential window is small, and the calculated specific capacitance will definitely be higher.

3. The manuscript only compared the volumetric capacitance of h-MoO₃ with α -MoO₃ and other electrode materials. However, the energy density and power density of the energy storage device are also important performance parameters. In this regard, the energy density and power density should also be compared in the manuscript.

4. The characterization and analyses of control sample MoO₃ are insufficient. For example, it has been reported that the structured water in MoO₃ is beneficial for proton migration. However, the manuscript only detected the water content of h-MoO₃ via TG curve. Did α -MoO₃ also contain structured water? In my opinion, the manuscript should also supplement

detection on the structured water in α -MoO₃.

5. More details on theoretical calculations should be supplemented. The manuscript calculated the proton transfer behavior in both h-MoO₃ and α -MoO₃. According to the computational details, the manuscript seems to only describe the model construction of h-MoO₃. What about α -MoO₃? Corresponding details should be supplemented.

6. For the structural model of MoO₃·0.7 H₂O used in theoretical calculations, the author put four H₂O molecules into MoO₃ unit cell. Is the constructed model consistent with the actual crystal model of MoO₃·0.7 H₂O? The water molecules in MoO₃·0.7 H₂O are crystal water molecules owning strong chemisorption interaction with MoO₃ crystal. However, the water molecules in theoretical model seem to only physically adsorb around MoO₃ unit according to Fig. 3. To confirm the reliability of constructed model, the author should compare its information on lattice structure (such as lattice parameter after water adsorption) with actual crystal structure.

7. According to the discussion in section Structural evolution of h-MoO₃, the proton intercalation and storage behavior are related to the exerted potential. According to the description of computational details, the AIMD simulation did not consider the electrochemical environment. In this case, can the performed AIMD simulation accurately describe the reaction and structure evolution process during charge and discharge? The author should supplement essential explanation for this point.

8. There are some typos in the manuscript. For example, "a-MoO₃" in line 138 should be corrected as " α -MoO₃". The author should check the manuscript more carefully.

We thank the reviewers for their time and very useful comments in improving the quality of this manuscript. Provided below is our detailed response to each question.

REVIEWER COMMENTS

Reviewer #1 (Remarks to the Author):

This work focused on the interesting finding for the fast proton transport phenomenon of a bulk hexagonal molybdenum oxide ($h\text{-MoO}_3$) with useful ion channels for energy storage. The topic is generally interesting to the potential readers in this journal but a major revision with considering below comments is required for the possible publication in this journal.

Comments 1: Introduction. From the literature, the electroactive materials with their charge storage mechanisms involving the exchange of protons have to be reviewed and compared in order to clarify whether the finding for the fast proton transport phenomenon of a bulk hexagonal molybdenum oxide ($h\text{-MoO}_3$) is new or not. Accordingly, the authors cannot ignore the typical references discussing the materials with the similar characteristics for pseudocapacitors. For example, RuO_2 in various hydrated degrees have been demonstrated to be the typical materials for pseudocapacitors with proton exchange (e.g., see J. Phys. Chem. B 106 (2002) 3592 and Chem. Mater. 19 (2007) 2112). In addition, the mixed electron and proton conductivity of hydrous RuO_2 have been discussed (Langmuir 15 (1999) 774, Angew. Chem. Int. Ed. 42 (2003) 4092, and J. Phys. Chem. B 109 (2005) 7330). The authors need to give a balanced report on the important references in the literature and compare/comment their idea with these typical references in order to emphasize and clarify their originality and novelty.

Our response: Thanks for the good question. According to this valuable suggestion, some typical electrode materials for proton storage have been reviewed and compared

in Introduction. Specifically, all references suggested by the reviewer were cited to review the development of RuO₂. As a typical pseudocapacitive material, the relationship between physical structure and electrochemical performance of RuO₂ was discussed, where the effect of nanostructure and structural water on the electrochemical performance was emphasized. However, these reported electrode materials (specially pseudocapacitive materials) for proton storage are minimized to nanometer-sized dimensions, leading to poor volumetric performance, serious side reactions, high cost, and complexity. It is critical to developing aqueous energy storage micrometer-scale bulk materials with fast chargeability and structural stability for large-scale energy storage. The realization of fast chargeability and structural stability without nanostructuring in bulk materials will have a breakthrough impact on conventional strategies and structural motifs for proton storage materials and pseudocapacitive materials. Please see Line 18, Page 4 to Line 11, Page 5 in Introduction, highlighted in yellow.

Comments 2: From the decay in capacitance with the scan rate or GCD current density, the capacitance retention is poor in comparison with the typical pseudocapacitor materials (e.g., Angew. Chem. Int. Ed. 42 (2003) 4092, Nano Lett. 6 (2006) 2690, and Electrochim. Acta 54 (2009) 4574 for RuO₂; J. Mater. Chem. A 5 (2017) 10021 and J. Electrochem. Soc. 166 (2019) A1875 for M_xMnO₂; J. Power Sources 196 (2011) 2387 for WO₃). The authors have to compare and comment this property with suitable references because of the supercapacitor application.

Our response: Thanks for the good question. Since the high-rate performance of electrodes relies critically on the transport kinetics of ions and electrons, most high-rate electrodes have mostly focused on the adoption of nano-sized or nanoarchitected materials. Nanostructuring increases the interfacial area between electrodes and the electrolytes and generates more reactive sites accessible for charge transfer reactions. Besides, the reduced transport path of ions in nanostructured materials is conducive to the adsorption/insertion of ions at a high scan rate or current density, leading to the

better rate performance of electrode materials with small particle size. However, there are also some challenges, such as poor volumetric performance, serious side reactions, high cost, and complexity, limiting the application of nanosized electrode materials. As an alternative approach to nanosizing, a micrometer-sized bulk *h*-MoO₃ electrode material with selective proton channels has been designed without nanoscaling, to provide fast and stable host structures for proton intercalation and storage. Owing to the size effect of electrode materials, the capacitance retention at high rates isn't as high as the well-designed nanostructured pseudocapacitive materials mentioned by the reviewer. Nevertheless, the micrometer-sized *h*-MoO₃ (particle size of tens of micrometers) electrodes still maintain an acceptable capacitance at high rates in comparison with typical pseudocapacitive nanostructured materials recently reported. As far as we know, there are few reported pseudocapacitive materials achieving excellent electrochemical performance in such large particle sizes. We have compared and commented the rate performance of *h*-MoO₃ electrodes with some typical capacitive materials reported in the last five years. **Please see Line 4-11, Page 9 and Supplementary Table 1, highlighted in yellow.** Furthermore, it is worth mentioning that micrometer-sized superstructure motifs of *h*-MoO₃ electrode provide stable host structures for proton intercalation and storage, resulting in a state-of-the-art volumetric capacitance (1750 F cm⁻³, which surpasses most reported electrode materials) and an ultra-stable cycle life (no capacitance drops after 10,000 cycles).

Supplementary Table 1. Electrochemical performance comparison of microstructured *h*-MoO₃ with some typical nanostructured pseudocapacitive materials reported in the last five years.

Electrode	Electrolyte	Capacitance (F g ⁻¹)	Rate (F g ⁻¹)	Ref.
G3DTF	1 M H ₂ SO ₄	400 (1 mV s ⁻¹)	113 (100 mV s ⁻¹)	1

Mo ₁₃₂ ⁻	1 M H ₂ SO ₄	65 (1 mV s ⁻¹)	~20 (100 mV s ⁻¹)	2
DTAB-EEG				
Cl-MXene	1 M H ₂ SO ₄	304 (2 mV s ⁻¹)	<100 (100 mV s ⁻¹)	3
TiNbC MXene	3 M H ₂ SO ₄	381 (2 mV s ⁻¹)	155 (100 mV s ⁻¹)	4
Nb@COF	0.1 M H ₂ SO ₄	367 (1 mV s ⁻¹)	165 (100 mV s ⁻¹)	5
Mn ₃ O ₄ NW	1 M Na ₂ SO ₄	301 (1 mV s ⁻¹)	~74 (50 mV s ⁻¹)	6
MoO ₃ @CNT	1 M Na ₂ SO ₄	281 (1 mV s ⁻¹)	75 (200 mV s ⁻¹)	7
MnO ₂ @CNT	1 M Na ₂ SO ₄	337 (1 mV s ⁻¹)	150 (200 mV s ⁻¹)	7
δ -MnO ₂ @ α -MnO ₂	1 M Na ₂ SO ₄	348 (5 mV s ⁻¹)	~200 (100 mV s ⁻¹)	8
V ₂ O ₅ NBs	1 M Na ₂ SO ₄	180 (5 mV s ⁻¹)	~100 (100 mV s ⁻¹)	8
Ti ₂ V _{0.9} Cr _{0.1} C ₂ T _x	1 M KOH	553 (2 mV s ⁻¹)	233 (100 mV s ⁻¹)	9
Li-V ₂ CT _x	1 M LiOH	386 (2 mV s ⁻¹)	139 (100 mV s ⁻¹)	10
h -MoO ₃	0.5 M H ₂ SO ₄	569 (2 mV s ⁻¹)	235 (100 mV s ⁻¹)	This work

Comments 3: For the high-rate performance, the scan range of 1-100 mV/s is acceptable for the supercapacitor application but not typically high in comparison with those using the very high scan rates of 500-1000 mV/s or even 10000 mV/s. In addition, the anodic and cathodic *b*-values significantly deviate from the ideal value, 1.0. Again, the authors have to compare and comment this property with suitable references

because of the micrometer structure of their materials or other reasons. In my opinion, the authors have to prepare and compare the charge storage characteristics of nanostructured *h*-MoO₃ with those of bulk *h*-MoO₃ in order to emphasize their unique micrometer structure.

Our response: Thanks for the good question. For electrodes with high-rate performance, solid-state diffusion/surface reaction of ions and electrons is usually the rate-limiting step, determining the rate capability of the supercapacitor. Nanostructured materials are extensively studied for more active area and shorter ion transport lengths, in order to design electrode materials with high-rate performance. Furthermore, nanometer-sized carbon additives as inactive electrode components are used to enhance the electronic conductivity of composite electrodes. However, nanostructuring has a negative impact on volumetric performance, serious side reactions, high cost, and complexity. To investigate the intrinsic pseudocapacitive properties, bulk *h*-MoO₃ without nanoscaling and carbon modification was studied, achieving outstanding volumetric capacitance and cycle life in this work. Nevertheless, the micrometer-sized *h*-MoO₃ (particle size of tens of micrometers) electrodes still maintain an acceptable capacitance at high rates in comparison with typical pseudocapacitive nanostructured materials recently reported. As far as we know, there are few reported pseudocapacitive materials achieving excellent electrochemical performance in such large particle sizes. We have compared and commented the rate performance of *h*-MoO₃ electrodes with some typical capacitive materials reported in the last five years. **Please see Line 4-11, Page 9 and Supplementary Table 1, highlighted in yellow.** Besides, the electrode polarization derived from size effects and poor electronic conductivity may make the *b*-values deviate from the ideal value, which significantly reduces the electrode kinetics. Nanostructured *h*-MoO₃ with a particle size of about 500 nm to 1 μm are prepared to reveal size effect and charge storage characteristic (**Supplementary Fig. 11**). Due to more active area and shorter ion transport lengths, nanostructured *h*-MoO₃ shows good specific capacitance (645 F g⁻¹ at 2 mV s⁻¹) and rate performance (306 F g⁻¹ at 100 mV s⁻¹) (**Supplementary Fig. 12**). However, the large specific surface of nanomaterials also provides more active sites for other side reactions (such as hydrogen evolution reaction),

leading to poor coulombic efficiency and cyclic stability (Supplementary Fig. 13).

Supplementary Table 1. Electrochemical performance comparison of microstructured h -MoO₃ with some typical nanostructured pseudocapacitive materials reported in the last five years.

Electrode	Electrolyte	Capacitance (F g ⁻¹)	Rate (F g ⁻¹)	Ref.
G3DTF	1 M H ₂ SO ₄	400 (1 mV s ⁻¹)	113 (100 mV s ⁻¹)	1
Mo ₁₃₂ ⁻	1 M H ₂ SO ₄	65 (1 mV s ⁻¹)	~20 (100 mV s ⁻¹)	2
DTAB-EEG				
Cl-MXene	1 M H ₂ SO ₄	304 (2 mV s ⁻¹)	<100 (100 mV s ⁻¹)	3
TiNbC MXene	3 M H ₂ SO ₄	381 (2 mV s ⁻¹)	155 (100 mV s ⁻¹)	4
Nb@COF	0.1 M H ₂ SO ₄	367 (1 mV s ⁻¹)	165 (100 mV s ⁻¹)	5
Mn ₃ O ₄ NW	1 M Na ₂ SO ₄	301 (1 mV s ⁻¹)	~74 (50 mV s ⁻¹)	6
MoO ₃ @CNT	1 M Na ₂ SO ₄	281 (1 mV s ⁻¹)	75 (200 mV s ⁻¹)	7
MnO ₂ @CNT	1 M Na ₂ SO ₄	337 (1 mV s ⁻¹)	150 (200 mV s ⁻¹)	7
δ -MnO ₂ @ α -MnO ₂	1 M Na ₂ SO ₄	348 (5 mV s ⁻¹)	~200 (100 mV s ⁻¹)	8
V ₂ O ₅ NBs	1 M Na ₂ SO ₄	180 (5 mV s ⁻¹)	~100 (100 mV s ⁻¹)	8
Ti ₂ V _{0.9} Cr _{0.1} C ₂ T _x	1 M KOH	553 (2 mV s ⁻¹)	233 (100 mV s ⁻¹)	9
Li-V ₂ CT _x	1 M LiOH	386 (2 mV s ⁻¹)	139 (100 mV s ⁻¹)	10

Supplementary Fig. 11 a-b SEM images of $h\text{-MoO}_3$ nanoparticles after mechanical milling.

Supplementary Fig. 12 a Capacitance of nanostructured $h\text{-MoO}_3$ electrodes at various scan rates. **b** GCD curves of nanostructured $h\text{-MoO}_3$ electrodes at different current densities.

Supplementary Fig. 13 Cycling stability performance for nanostructured $h\text{-MoO}_3$ electrodes at 20 A g^{-1} .

Comments 4: The authors have to compare the CVs measured in the aqueous media with various pH values in order to confirm the charge storage mechanism of $h\text{-MoO}_3$ (e.g., pH from ca. 0 to 3). In other words, the redox potential of $h\text{-MoO}_3$ should be pH-dependent. In addition, water is a proton acceptor and donor, why the CV curves of $h\text{-MoO}_3$ electrodes measured in neutral electrolytes do not show the pseudocapacitive responses in a certain degree (e.g., half of the charges) if water molecules in ion channels can build a continuous hydrogen bond network for the ultra-fast synergistic transport of protons? RuO_2 and IrO_2 with the mixed electron and proton conductivity have been found to possess the redox pseudocapacitance in aqueous media with a very wide pH range (ca. pH from 0 to 14, e.g., see J. Electrochem. Soc. 139 (1992) 2158). The authors have to comment the charge storage characteristics of $h\text{-MoO}_3$ in comparison with RuO_2 and IrO_2 .

Our response: Thanks for the good question. The CVs of $h\text{-MoO}_3$ were measured in H_2SO_4 with various pH values at a scan rate of 2 mV s^{-1} (e.g., pH from ca. 0 to 3). As **Supplementary Fig. 7** shows, the oxidation peak shifts to higher potential and the reduction peak shifts to lower potential, demonstrating that the redox potential of $h\text{-MoO}_3$ is pH-dependent. This phenomenon can also be explained theoretically, according to the electrochemical reaction in the charging process and Nernst's equation:

$$\varphi = \varphi^\theta - \frac{RT}{0.66F} \ln[\text{H}^+] \quad (2)$$

During the oxidation process, the chemical potential (φ) was increased significantly with the reduced concentration of proton (H^+). Additionally, as the proton concentration decreases, the lower specific capacitance of *h*-MoO₃ is observed. When the pH value of the H₂SO₄ solution is ≥ 3 , an interesting feature shows negligible current response, which is similar to the electrochemical behavior in various neutral electrolytes (Supplementary Fig. 14). On the one hand, the decrease of active materials involved in the electrochemical reaction with the reduced number of protons causes a decrease in the specific capacitance. On the other hand, ion channels in *h*-MoO₃ are selective for protons due to the minimal physical size of protons and the diffusion-free topochemistry proton transport, resulting in that there is negligible pseudocapacitive response in neutral electrolytes. Please see Line 16-22, Page 8, Line 4-14, Page 12, Supplementary Fig. 7, and Supplementary Fig. 14, highlighted in yellow.

Following previously reported literature (e.g., Angew. Chem. Int. Ed. 42 (2003) 4092, J. Phys. Chem. C 117 (2013) 12003), RuO₂ and IrO₂ are typical pseudocapacitive materials based on surface redox pseudocapacitance mechanism, in which electrochemical performance of RuO₂ and IrO₂ depends on the particle size and specific surface area of active materials. On the contrary, the intercalation pseudocapacitance mechanism is revealed *in situ* X-ray diffraction (Figure 4) and theoretical calculation (Figure 5) in *h*-MoO₃, where the intercalation/de-intercalation of protons will cause the expansion and shrinkage of the crystal lattice. The illustration of redox pseudocapacitance mechanism and intercalation pseudocapacitance mechanism is given in Figure R1. As the reviewer noted, RuO₂ and IrO₂ possess redox pseudocapacitance in aqueous media with a very wide pH range. It is worth noting that nanoparticles with small sizes (diameter below tens of nanometers) are used to improve the electrochemical performance of RuO₂. The ultrahigh specific surface area of nanoparticles may provide abundant active sites for surface oxidation reactions of other charge carriers and generate extrinsic pseudocapacitance, leading to similar redox pseudocapacitance in aqueous media. For micrometer-sized *h*-MoO₃, the specific capacitance in neutral electrolytes is hindered by the low active surface. Finally, these pseudocapacitive materials (*h*-MoO₃, RuO₂, and IrO₂) for proton storage are aided by

crystal water providing proton transport paths and small-size particles ensuring sufficient binding sites. Please see Line 15-19, Page 18, highlighted in yellow.

Supplementary Fig. 7 CV curves of *h*-MoO₃ electrode in H₂SO₄ with various pH values at a scan rate of 2 mV s⁻¹.

Supplementary Fig. 14 CV curves of *h*-MoO₃ electrodes at different electrolytes at a scan rate of 5 mV s⁻¹.

Figure R1. **a** Illustration of redox pseudocapacitance. **b** Illustration of intercalation pseudocapacitance.

Comments 5: What is the gravity-based specific capacitance of $h\text{-MoO}_3$? From the CVs of $h\text{-MoO}_3$, the gravity-based specific capacitance of $h\text{-MoO}_3$ is similar to that of WO_3 . However, the density of WO_3 is ca. 7.16 g cm^{-3} , indicating that the volumetric capacitance of WO_3 (e.g., see *J. Power Sources* 196 (2011) 2387 and *Electrochem. Commun.* 12 (2010) 1800) is higher than that of $h\text{-MoO}_3$. The authors have to compare and comment this concern in Fig. 2d.

Our response: Thanks for the good question. The gravity-based specific capacitance of $h\text{-MoO}_3$ has been added to the Fig. 2c. The $h\text{-MoO}_3$ electrode delivers a maximum specific capacitance of 569 F g^{-1} at a scan rate of 2 mV s^{-1} as well as a specific capacitance of 235 F g^{-1} at a scan rate of 100 mV s^{-1} . Because W and Mo are in group VI B of the periodic table, $h\text{-MoO}_3$ and WO_3 exhibit similar pseudocapacitive behavior in H_2SO_4 . The high intrinsic density of MoO_3 and WO_3 is promising for the design of electrode materials and devices with excellent volumetric properties. To enhance the electrochemical performance of WO_3 electrodes, many nanostructuring strategies have been used to design various nanostructured WO_3 . (e.g., see *J. Power Sources* 196 (2011) 2387 and *Electrochem. Commun.* 12 (2010) 1800) However, the void space between nanometer-sized materials reduces the tap density of particles, resulting in that the volumetric capacitance of nanostructured WO_3 tends to be poor. The volumetric capacitance of typical WO_3 materials has been compared and commented in Manuscript

and Fig. 2d, where a more competitive volumetric capacitance is achieved in *h*-MoO₃ electrodes (1750 F cm⁻³) instead of WO₃ electrodes (340 F cm⁻³).

Comments 6: The degradation mechanism of alpha-MoO₃ needs to be confirmed by direct evidences via various material characterization tools.

Our response: Thanks for the good question. The structural evolution and degradation mechanism of α -MoO₃ electrodes is evaluated using *in situ* XRD measurements. During the initial discharge, the new diffraction peaks at 50–55° appear, while some diffraction peaks (such as (110)) almost disappear, suggesting the formation of a new phase H_xMoO₃ (Supplementary Fig. 22a-c). During charging and the following discharging, a typical new/old phase transition reaction occurs, accompanying the intercalation/deintercalation of protons in α -MoO₃ (Supplementary Fig. 22d-e). Particularly, the obvious diffraction peak shift during the electrochemical reaction is associated with the strong electrostatic interaction between charge carriers and electrode hosts, leading to huge internal strain and the structure collapse of electrodes. Another reason is that the active Mo⁶⁺ would react with water molecules in the electrolyte and generate soluble Mo-O oligomer species, resulting in Mo dissolution and surface structure degradation (see Mater. Res. Bull. 31 (1996) 1537). The durability of electrodes was demonstrated by ICP results in Supplementary Table 2, where the Mo content in the electrolyte for *h*-MoO₃ and α -MoO₃ after 1000 cycles is 69.9 and 123 mg L⁻¹, respectively.

Supplementary Fig. 22 Structural evolution of α -MoO₃. **a** The first discharge curve of α -MoO₃ (a current density of 0.03 A g⁻¹). **b** Contour plots of *in situ* XRD during the first discharge process. **c** *In situ* XRD patterns of different diffraction peaks during the first discharge. **d** GCD curves of α -MoO₃ (the first charge and second discharge processes). **e** *In situ* XRD patterns during charge and discharge processes. **f** *In situ* XRD pattern of different diffraction peak during charge and discharge processes.

Supplementary Table 2. ICP-OES results of electrolytes for *h*-MoO₃ and α -MoO₃ after 1000 cycles.

electrode	h -MoO ₃	α -MoO ₃
Mo content (mg L ⁻¹)	69.9	123

Comments 7: Do the authors make sure that *h*-MoO₃ can proceed the redox reactions if Li⁺, Na⁺, or K⁺ can exchange with *h*-MoO₃? In my opinion, if the redox reactions of *h*-MoO₃ do occur with the exchange of Li⁺, Na⁺, or K⁺, the active sites on the superficial region should contribute the pseudocapacitance in the neutral media. How about the authors opinions? The authors should give their opinions in the revision or supporting materials.

Our response: Thanks for the good question. Cyclic voltammetry (CV) curves for *h*-MoO₃ electrodes are carried out in electrolytes with different charge carriers (such as Li₂SO₄, Na₂SO₄, K₂SO₄, and (NH₄)₂SO₄) (Supplementary Fig. 14). All CV curves show negligible current response, suggesting that the electrochemical intercalation of the above ions doesn't occur in bulk *h*-MoO₃ (the H⁺ intercalation pseudocapacitance mechanism is revealed by *in situ* X-ray diffraction (Figure 4) and theoretical calculation (Figure 5) in *h*-MoO₃). The characteristic of tiny electrical double-layer capacitance shows the absence of significant pseudocapacitance, which may be associated with the low active surface in micrometer-sized *h*-MoO₃. On the one hand, protons have the smallest ionic radius and mass that are favorable for fast transport in these channels, while Li⁺, Na⁺, K⁺, and NH₄⁺ with larger ionic radius and masses are more difficult to intercalate into electrodes. On the other hand, water molecules in ion channels can build a continuous hydrogen bond network for the ultra-fast synergistic transport of protons and water molecules (Li⁺, Na⁺, and K⁺ might only be transported by slow ionic diffusion). The opinions are described in Line 4-14, Page 12 and Supplementary Fig. 14, highlighted in yellow.

Supplementary Fig. 14 CV curves of *h*-MoO₃ electrodes at different electrolytes at a scan rate of 5 mV s⁻¹.

Comments 8: What is the definition of bulk resistance? How to obtain the bulk

resistance from the EIS data? For EIS data, in the high frequency region (e.g., > 2.5 kHz), mass transfer can be neglected and the charge-transfer process at the outermost surface of the electrode in contact with the electrolyte will be dominant. This behavior is typical for pseudocapacitor materials. Why did the semicircle in the high frequency region represent the bulk impedance which indicates the transport rate of protons in the h -MoO₃ bulk phase?

Our response: Thanks for the good question. Based on previous literature, a coin cell with h -MoO₃ powder as the solid electrolyte was assembled to test the electrochemical impedance spectroscopy (EIS). Under the electric field, proton migration within the h -MoO₃ powder leads to the resistance of the bulk powder (bulk resistance) (Supplementary Fig. 16). Besides, the grain boundary resistance shows proton migration at the grain boundaries where the particles are in contact with each other under pressure stacking. In solid-state electrolytes, the mass transport of ions will dominate the total resistance in the high frequency region. The semicircle in the high frequency region corresponds to the bulk resistance of the solid electrolyte, which is different from the EIS of typical pseudocapacitive materials in liquid electrolytes. The proton conductivity was obtained according to the conductivity equation, indicating the transport rate of protons in bulk h -MoO₃. (e.g., see Nat. Energy 4 (2019) 123, ACS Energy Lett. 4 (2019) 2805, and Sci. Adv. 9 (2023) eadf4589.)

Supplementary Fig. 16 The electrochemical impedance spectrum of h -MoO₃ at different temperature (Inset is the equivalent circuit).

Comments 9: The authors have to provide the equivalent circuit for fitting the EIS data and clarify all the elements in the equivalent circuit (e.g., see J. Phys. Chem. B 109 (2005) 7330). Note that the frequency was swept from 7 MHz to 100 mHz in this work; why did the impedance obtained at the high-frequency end not approach the real part axis in Fig. 3a? For most Faradaic reactions, the impedance obtained at the high-frequency end will approach the real part axis at ca. 50kHz–1MHz (e.g., see Supplementary Fig. 9).

Our response: Thanks for the good question. In this paper, a coin cell with *h*-MoO₃ powder as solid electrolytes was assembled to test the electrochemical impedance spectroscopy (EIS). The equivalent circuit has been added in **Supplementary Fig. 16**, where R_b and R_{gb} represent the resistance of the bulk power and their grain boundary, respectively. The bulk response frequency for many inorganic solid electrolytes is extremely high and exceeds the highest frequency that can be measured by the electrochemical workstation. Therefore, a portion of the semicircle can only be observed in the high frequency region, where the bulk resistance can be obtained by fitting the semicircle. (e.g., see J. Am. Chem. Soc. 132 (2010) 6620, Nat. Energy 4 (2019) 123.) Some similar EIS spectrums have been reported in electrolytes of solid-state batteries. (e.g., see Adv. Energy Mater. 10 (2020) 2001497, ACS Appl. Mater. Interfaces 12 (2020) 56118)

Supplementary Fig. 16 The electrochemical impedance spectrum of *h*-MoO₃ at

different temperature (Inset is the equivalent circuit).

Comments 10: The water content of $h\text{-MoO}_3 \cdot 0.7\text{H}_2\text{O}$ is very high ($\text{Mo}:\text{H}_2\text{O} = 1:0.7$). In addition, I did not see the suitable ratio between Mo and water from the models shown in Supplementary Fig. 10 and 11. In addition, from the model shown in Supplementary Fig. 12, I cannot find the role of hydrated water favoring the transport of proton. Accordingly, the models proposed here are too arbitrary to be acceptable.

Our response: Thanks for the good question. The water content of $h\text{-MoO}_3 \cdot 0.7\text{H}_2\text{O}$ was determined and optimized based on physical characterizations and density functional theory (DFT), which is similar to the reported $\text{WO}_3 \cdot n\text{H}_2\text{O}$. To show the proton transport and storage process more clearly, some local image of internal tunnels of the actual crystal model $\text{MoO}_3 \cdot 0.7\text{H}_2\text{O}$ is provided. Following the reviewer's request, we have added the models of $h\text{-MoO}_3 \cdot 0.7\text{H}_2\text{O}$ with the suitable ratio between Mo and water in Supplementary Fig. 18 and 19. We have added more characterization and analyses of $\alpha\text{-MoO}_3$, such as Raman spectra and FTIR spectra in Supplementary Fig. 1 and TG curves in Supplementary Fig. 5. The absence of the characteristic peaks for O-H illustrates the absence of structural water, which is attributed to the narrow lattice space in $\alpha\text{-MoO}_3$. The TG curve shows the ignorable mass loss in $\alpha\text{-MoO}_3$ with increasing temperature, suggesting that there is no crystal water in $\alpha\text{-MoO}_3$. Based on the physical characterization and theoretical calculations, we have rebuilt and optimized the models in Supplementary Fig. 20 and other models.

Supplementary Fig. 18 Structural schematic of possible binding sites for protons in $h\text{-MoO}_3 \cdot 0.7\text{H}_2\text{O}$.

MoO₃·0.7H₂O. (a) A view down the c direction of the structure. (b) A view down the b direction of the structure.

Supplementary Fig. 19. The binding sites of protons in *h*-MoO₃·0.7H₂O.

Supplementary Fig. 20. The migration pathway of protons in α -MoO₃.

Comments 11: The asymmetric device exhibits a maximum energy density of 36 Wh/kg at a power density of 1.0 kW/kg, and remains 21 Wh/kg at 21 kW/kg. The performance is good but not very outstanding since most Ni-Co oxyhydroxides/sulfides-based asymmetric/hybrid cells (e.g., see J. Power Sources 253 (2014) 205) can provide the similar or even better performance than the cell demonstrated in this work. The authors need to make correct comparisons and clarify their merits of their material and cell designs from the typical references.

Our response: Thanks for the good question. Taking advantage of the high capacity and rapid kinetics of proton topochemistry in bulk *h*-MoO₃ engenders an exciting avenue for breaking through the power limit of batteries and the energy limit of

capacitors. The proton pseudocapacitor based on *h*-MoO₃ electrodes exhibits a maximum energy density of 36 Wh kg⁻¹ at a power density of 1.0 kW kg⁻¹, and remains a high energy density of 21 Wh kg⁻¹ at a maximum power density of 21 kW kg⁻¹, which will maximize the energy and power density of supercapacitors. The higher energy density is achieved at a high rate, demonstrating its potential value for high-power device applications. The electrochemical performance of this proton pseudocapacitor (especially the energy density at maximum power density) surpasses not only most supercapacitors in acid electrolytes, but also many typical supercapacitors in alkaline and neutral electrolytes (Supplementary Table 4), such as 2-GCE//AC (23.4 Wh kg⁻¹ at 3.9 kW kg⁻¹ and 21.1 kW kg⁻¹ at 10 Wh kg⁻¹) and 1@Ti₃C₂T_x//ACL (32.2 Wh kg⁻¹ at 2.4 kW kg⁻¹ and 12 kW kg⁻¹ at 19.7 Wh kg⁻¹) in acid electrolytes, MLMO//MLMO (34.1 Wh kg⁻¹ at 0.8 kW kg⁻¹ and 10 kW kg⁻¹ at 14.5 Wh kg⁻¹) in alkaline electrolytes, and MoO₃@CNT//MnO₂@CNT (27.8 Wh kg⁻¹ at 0.52 kW kg⁻¹ and 10 kW kg⁻¹ at 8 Wh kg⁻¹) in neutral electrolytes. Please see Line 11-22, Page 19 and Supplementary Table 4, highlighted in yellow.

Supplementary Table 4. Electrochemical performance comparison of our work with various supercapacitors reported in the last five years.

Electrode	Electrolyte	Max Energy density (Wh kg ⁻¹)	Max power density (kW kg ⁻¹)	Ref.
2-GCE//AC	3 M H ₂ SO ₄	23.4 (3.9 kW kg ⁻¹)	21.1 (10 Wh kg ⁻¹)	47
RuO ₂ //Hex-Aza-COF- 3	1 M H ₂ SO ₄	23.3 (0.67 kW kg ⁻¹)	—	48
1@Ti ₃ C ₂ T _x //ACL	2 M H ₃ PO ₄	32.2 (2.4 kW kg ⁻¹)	12 (19.7 Wh kg ⁻¹)	49
W ₁₈ O ₄₉ /Ti ₃ C ₂ T _x //RuO 2@CC	1 M H ₂ SO ₄	29.6 (0.75 kW kg ⁻¹)	7 (24.1 Wh kg ⁻¹)	50
MWCNT/RuO ₂ //WO ₃	1 M H ₂ SO ₄	27.2 (0.75 kW kg ⁻¹)	—	51

$/\text{Ti}_3\text{C}_2\text{T}_x$				
AC//2-CPE	0.5 M H_2SO_4	16.1 (1.7 kW kg^{-1})	10 (10.8 Wh kg^{-1})	52
PYT/GN 4-5//A-	1 M H_2SO_4	18.4 (0.7 kW kg^{-1})	7 (13.2 Wh kg^{-1})	53
$\text{Ti}_3\text{C}_2\text{T}_x$				
Ni-S/1d- Ti_3C_2 //d-	6 M KOH	20 (0.5 kW kg^{-1})	10 (5.1 Wh kg^{-1})	54
Ti_3C_2				
STO//STO	3 M KOH	27.8 (0.3 kW kg^{-1})	19.2 (—)	55
MLMO//MLMO	3 M KOH	34.1 (0.8 kW kg^{-1})	10 (14.5 Wh kg^{-1})	56
MoO_3 @CNT// MnO_2	1 M Na_2SO_4	27.8 (0.52 kW kg^{-1})	10 (9.8 Wh kg^{-1})	7
@CNT				
A-Ni-	1 M Na_2SO_4	27.8 (0.49 kW kg^{-1})	11 (9 Wh kg^{-1})	57
MnBMO//FCNT				
$\text{Cu}_{0.82}\text{Co}_{0.18}\text{HCF}$ //	0.5 M H_2SO_4	36 (1 kW kg^{-1})	21 (21 kW kg^{-1})	This
$h\text{-MoO}_3$				work

Reviewer #2 (Remarks to the Author):

The authors insist that a bulk hexagonal molybdenum oxide (MoO_3) with ion channels enables the fast transport of protons due to its having hydrogen bonding topochemistry, as demonstrated by several orders of magnitude higher than those of orthorhombic MoO_3 . Besides, it achieves an volumetric capacitance ($\sim 1750 \text{ F/cm}^3$) and long cycle life ($>10,000$ cycles). They also tried to unveil the mechanism through in-situ XRD and density functional theory calculations.

Comments 1: The synthesized methods for $h\text{-MoO}_3$ with open channels appear to have been already reported by References 25 and 26 so that this work is considered to report the electrochemical performances of $h\text{-MoO}_3$ with open channels originally reported through the previous studies. This work is interesting in views of volumetric capacitance, rate-capability, and cycle life, but its electrode structure is simply based on $h\text{-MoO}_3$ (similar to the previous studies) whose open channels are commonly expected to enable fast transports of protons and other ions leading to high power density as well as cycle stability. Furthermore, the high tap density of the hexagonal microrods is expected from the morphology from $h\text{-MoO}_3$.

Of course, the finding of $h\text{-MoO}_3$ with adsorbed waters to enhance proton transports appears to be very interesting, although similar mechanisms (For example, Reference 23) have been already reported.

Our response: Thanks for the good question. As the reviewer noted, a few traditional nanostructured $h\text{-MoO}_3$ have been reported for electrochemical energy storage. Unfortunately, nanostructured materials still have introduced fundamental challenges, including poor volumetric performance, serious side reactions, high cost, and complexity. Based on References 25 and 26, a micrometer-sized bulk $h\text{-MoO}_3$ (a diameter of about 2–4 μm and a length of 10–30 μm) is designed without nanoscaling by adjusting the experimental process, which breaks the conventional nanostructuring strategy of electrode materials. For the first time, microstructured $h\text{-MoO}_3$ microrods we reported are found to deliver a higher specific capacitance of 569 F g^{-1} at a potential

window of -0.53 to 0 V, indicating a more suitable anode material for fast and stable proton storage. Specifically, micrometer-sized superstructure motifs of *h*-MoO₃ electrode provide stable host structures for proton intercalation and storage, resulting in a state-of-the-art volumetric capacitance (1750 F cm⁻³, which surpasses most reported electrode materials) and an ultra-stable cycle life (no capacitance drops after 10,000 cycles). The diffusion-free proton transport kinetics based on hydrogen bonding topochemistry is demonstrated in *h*-MoO₃ whose proton conductivity is several orders of magnitude higher than traditional orthorhombic molybdenum oxide (α -MoO₃). Theoretical calculation and *in situ* X-ray diffraction technology show that protonation-induced structural transformation optimizes the crystal structure of *h*-MoO₃ electrodes and realizes rapid reversible solid-state proton transport and storage. This work offers novel insights into electrode structure design principles for bulk materials with fast chargeability and structural stability, and brings light to the transport and storage mechanism of non-metallic charge carriers.

Comments 2: Additionally, the authors did not describe the mass loadings (both three-electrode and also full-cell configurations) for their electrochemical performances, which is important to judge the novelty of performances. It is notable that electrochemical performances vary depending on different mass loadings. Whether their electrode could be practically usable as a commercial-type, their performances should be really demonstrated at high mass loadings.

Our response: Thanks for the good question. Several *h*-MoO₃ electrodes with the active mass of 3, 5, and 10 mg cm⁻² are investigated for the practical application of electrode materials. The volumetric capacitance slightly decreases with increased active mass, but remains at 813 F cm⁻³ even under 10 mg cm⁻² (Supplementary Fig. 15a). The rate-dependent fading tendency also occurs with the increase of the *h*-MoO₃ mass loading. For pursuing ultrahigh areal capacitance towards miniaturized devices, the areal capacitance of *h*-MoO₃ electrodes at 1 mV s⁻¹ is 1245, 1707, and 2622 mF cm⁻², respectively (Supplementary Fig. 15b). When increasing the scan to 20 mV s⁻¹, the area capacitance of over 600 mF cm⁻² is achieved in high-mass loading, which is superior

to activated carbon with an acceptable area capacitance of 0.6 F cm^{-2} . Bulk $h\text{-MoO}_3$ electrode materials show potential value for designing thick electrodes with exceptional electrochemical performance and constructing supercapacitors with high volumetric and areal energy densities.

Supplementary Figure 10. (a) Volumetric capacitance of $h\text{-MoO}_3$ electrodes at different mass loadings. (b) Areal capacitance of $h\text{-MoO}_3$ electrodes at different mass loadings.

Comments 3: Besides, the generalized gradient approximation (GGA) based on Perdew-Burke-Ernzerhof (PBE) exchange functional functional is subjected to the underestimation of kinetic energies (over-performing). Especially, proton transfer using PBE exchange could be overestimated using PEB. Also, the authors did not specify which correlation functional is used in their DFT calculations. Another issue is relating to their energies. The authors need to clarify whether their energies include vibration energies and thermal corrections or not.

Our response: Thanks for the good question. The underestimation of kinetic energies in GGAs like PBE can be attributed to the lack of explicit treatment of electron correlation effects. To address the limitations of GGA, meta-GGA has been used to provide better accuracy for a wide range of properties, including kinetic energies. Meta-GGA is used to verify the vacancy formation energy. The results obtained by meta-GGA are in agreement with the GGA results, demonstrating the applicability of meta-GGA and GGA in this system. Additionally, the PBE exchange functional was used to carry out AIMD simulations, while the formation energy of H vacancy was calculated

by meta-GGA exchange functional. Besides, the energies of DFT calculations in this paper don't include vibration energies and thermal corrections. To more accurately describe the reaction and structure evolution process, the proton intercalation and storage behavior related to the exerted potential was also considered. MD simulations and DFT calculations in different electrochemical environment are performed, where $H_{0.5}MoO_3$, $HMoO_3$, and H_2MoO_3 (different H contents correspond to different potentials) occur in electrochemical reaction stage I, II, and III in Fig 4a, respectively (Supplementary Fig. 21 and Fig. 5b). Based on AIMD simulations, protons are intercalated into the electrode bulk phase to generate successively three compositions during the first discharge process. In stage I, protons will combine with crystal waters to form hydronium— H_3O^+ . In stage II, H bonds further with H to form H-H and fills in hexagonal tunnels. In stage III, oxygen atoms of the MoO_6 octahedra can be occupied, accompanied by H bonding to O as O-H to form H_xMoO_3 (Fig. 5a). The binding energy of formed H_2 and H_3O^+ is too high to kick out free protons in the subsequent charging process (Fig. 5c), leading to 1.34 mol hydrogen ions permanently occupying the internal space of $h-H_{1.34}MoO_3 \cdot 0.7H_2O$. $h-H_2MoO_3 \cdot 0.7H_2O$ undergoes a structural self-optimization process to generate 0.66 mol protons acted as charge carriers. Then after the electrode is charged to 0 V, the protons binding with H_xMoO_3 (about 0.66 mol protons) can be de-intercalated from $h-H_2MoO_3 \cdot 0.7H_2O$ to form $h-H_{1.34}MoO_3 \cdot 0.7H_2O$, and are reversibly inserted/de-inserted during following charging/discharging reaction to achieve outstanding structural stability. The first discharge process is as follows:

In the subsequent charging and discharging process, the reaction mechanism is described according to the following equation:

Supplementary Figure 21. **a** The distribution of H coordination environments from the room temperature AIMD simulations of $h\text{-H}_{0.5}\text{MoO}_3\cdot 0.7\text{H}_2\text{O}$ structure. **b** The distribution of H coordination environments from the room temperature AIMD simulations of $h\text{-HMoO}_3\cdot 0.7\text{H}_2\text{O}$ structure.

Fig. 5 Energy storage mechanism of $h\text{-MoO}_3$. **a** Schematic depiction of the energy storage mechanism in $h\text{-MoO}_3\cdot 0.7\text{H}_2\text{O}$. **b** The distribution of H coordination environments from the room temperature AIMD simulations of $h\text{-H}_2\text{MoO}_3\cdot 0.7\text{H}_2\text{O}$ structure. **c** The formation energies of H vacancy with three different H coordination

environments of *h*-H₂MoO₃·0.7H₂O structure.

Reviewer #3 (Remarks to the Author):

The manuscript reported a type of bulk electrode material with both excellent volumetric capacitance for proton storage and cycling stability as compared with previously reported nanostructured material. There are some concerns to be addressed before its possible publication:

Comments 1: *h*-MoO₃ have been reported for pseudocapacitive energy storage with proton or NH₄⁺ as charge carrier through Grotthuss or Vehicle mechanism (Nanoscale 2015, 7, 11777; Adv. Mater. 2020, 32, 190780). The preparation method of this work is also based on Adv. Mater. 2020, 32, 190780.

Our response: Thanks for the good question. As the reviewer noted, a few traditional nanostructured *h*-MoO₃ have been reported for pseudocapacitive energy storage with proton or NH₄⁺ as charge carriers. Unfortunately, nanostructured materials still have introduced fundamental challenges, including poor volumetric performance, serious side reactions, high cost, and complexity. Based on References 25 and 26, a micrometer-sized bulk *h*-MoO₃ (a diameter of about 2–4 μm and a length of 10–30 μm) is designed without nanoscaling by adjusting the experimental process, which breaks the conventional nanostructuring strategy of electrode materials. While nanostructured *h*-MoO₃ pyramidal nanorods only achieve a specific capacitance of about 230 F g⁻¹ at a potential window of 0.05 to 0.65 V (Nanoscale 2015, 7, 11777), microstructured *h*-MoO₃ microrods we reported are firstly found to deliver a higher specific capacitance of 569 F g⁻¹ at a potential window of -0.53 to 0 V (This work), indicating a more suitable anode material for proton storage. Specifically, micrometer-sized superstructure motifs of *h*-MoO₃ electrodes provide stable host structures for proton intercalation and storage, resulting in a state-of-the-art volumetric capacitance (1750 F cm⁻³, which surpasses most reported electrode materials) and an ultra-stable cycle life (no capacitance drops after 10,000 cycles). The diffusion-free proton transport kinetics based on hydrogen bonding topochemistry is demonstrated in *h*-MoO₃ whose proton conductivity is several orders of magnitude higher than traditional orthorhombic molybdenum oxide

(α -MoO₃). Theoretical calculation and *in situ* X-ray diffraction technology show that protonation-induced structural transformation optimizes the crystal structure of *h*-MoO₃ electrodes and realizes rapid reversible solid-state proton transport and storage. This work offers novel insights into electrode structure design principles for bulk materials with fast chargeability and structural stability, and brings light to the transport and storage mechanism of non-metallic charge carriers.

Comments 2: As for the high volumetric performance, the calculation details of volumetric capacitance should be given. Why the author choose the potential window of -0.5 V to 0 V in three-electrode test? Compared to many literatures, the applied potential window is small, and the calculated specific capacitance will definitely be higher.

Our response: Thanks for the good question. The calculation details of volumetric capacitance are given in Calculation. The packing density (ρ) was measured by determining the geometrical parameters and total electrode mass after *h*-MoO₃ powder without the addition of conductive carbon was coated on the current collector. Then volumetric capacitance (C_v) was calculated by using the following equation:

$$C_v = \rho C_g \quad (9)$$

The CV curves of *h*-MoO₃ electrodes at a potential window of -0.6 to 0.2 V are given in **Figure R2**. When the potential is extended to -0.6 V, excess voltage accelerates hydrogen evolution reaction (HER) on the interface between electrodes and electrolytes, which poses detrimental consequences toward the cyclic life. In the potential window of 0 to 0.2 V, the small-area CV curves imply poor charge response. To get better charge storage performance and stability of electrode structure, we have to choose the potential window of -0.53 V to 0 V in the three-electrode test. The similar applied potential window is used in other anode materials (ACS Appl. Mater. Interfaces 2014, 6, 18901; Nat. Nanotech., 2021, 17, 153).

Fig. R2 CV curves at a potential window of -0.6 to 0.2 V

Comments 3: The manuscript only compared the volumetric capacitance of *h*-MoO₃ with α -MoO₃ and other electrode materials. However, the energy density and power density of the energy storage device are also important performance parameters. In this regard, the energy density and power density should also be compared in the manuscript.

Our response: Thanks for the good question. Taking advantage of the high capacity and rapid kinetics of proton topochemistry in bulk *h*-MoO₃ engenders an exciting avenue for breaking through the power limit of batteries and the energy limit of capacitors. The proton pseudocapacitor based on *h*-MoO₃ electrodes exhibits a maximum energy density of 36 Wh kg⁻¹ at a power density of 1.0 kW kg⁻¹, and remains a high energy density of 21 Wh kg⁻¹ at a maximum power density of 21 kW kg⁻¹, which will maximize the energy and power density of supercapacitors. The higher energy density is achieved at a high rate, demonstrating its potential value for high-power device applications. The electrochemical performance of this proton pseudocapacitor (especially the energy density at maximum power density) surpasses not only most supercapacitors in acid electrolytes, but also many typical supercapacitors in alkaline and neutral electrolytes (Supplementary Table 4), such as 2-GCE//AC (23.4 Wh kg⁻¹ at 3.9 kW kg⁻¹ and 21.1 kW kg⁻¹ at 10 Wh kg⁻¹) and 1@Ti₃C₂T_x//ACL (32.2 Wh kg⁻¹ at 2.4 kW kg⁻¹ and 12 kW kg⁻¹ at 19.7 Wh kg⁻¹) in acid electrolytes, MLMO//MLMO (34.1 Wh kg⁻¹ at 0.8 kW kg⁻¹ and 10 kW kg⁻¹ at 14.5 Wh kg⁻¹) in alkaline electrolytes,

and MoO₃@CNT//MnO₂@CNT (27.8 Wh kg⁻¹ at 0.52 kW kg⁻¹ and 10 kW kg⁻¹ at 8 Wh kg⁻¹) in neutral electrolytes. Please see Line 11-22, Page 19 and Supplementary Table 4, highlighted in yellow.

Supplementary Table 4. Electrochemical performance comparison of our work with various supercapacitors reported in the last five years.

Electrode	Electrolyte	Max Energy density (Wh kg ⁻¹)	Max power density (kW kg ⁻¹)	Ref.
2-GCE//AC	3 M H ₂ SO ₄	23.4 (3.9 kW kg ⁻¹)	21.1 (10 Wh kg ⁻¹)	47
RuO ₂ //Hex-Aza-COF- 3	1 M H ₂ SO ₄	23.3 (0.67 kW kg ⁻¹)	—	48
1@Ti ₃ C ₂ T _x //ACL	2 M H ₃ PO ₄	32.2 (2.4 kW kg ⁻¹)	12 (19.7 Wh kg ⁻¹)	49
W ₁₈ O ₄₉ /Ti ₃ C ₂ T _x //RuO 2@CC	1 M H ₂ SO ₄	29.6 (0.75 kW kg ⁻¹)	7 (24.1 Wh kg ⁻¹)	50
MWCNT/RuO ₂ //WO ₃ /Ti ₃ C ₂ T _x	1 M H ₂ SO ₄	27.2 (0.75 kW kg ⁻¹)	—	51
AC//2-CPE	0.5 M H ₂ SO ₄	16.1 (1.7 kW kg ⁻¹)	10 (10.8 Wh kg ⁻¹)	52
PYT/GN 4–5//A- Ti ₃ C ₂ T _x	1 M H ₂ SO ₄	18.4 (0.7 kW kg ⁻¹)	7 (13.2 Wh kg ⁻¹)	53
Ni–S/1d-Ti ₃ C ₂ //d- Ti ₃ C ₂	6 M KOH	20 (0.5 kW kg ⁻¹)	10 (5.1 Wh kg ⁻¹)	54
STO//STO	3 M KOH	27.8 (0.3 kW kg ⁻¹)	19.2 (—)	55
MLMO//MLMO	3 M KOH	34.1 (0.8 kW kg ⁻¹)	10 (14.5 Wh kg ⁻¹)	56
MoO ₃ @CNT//MnO ₂	1 M Na ₂ SO ₄	27.8 (0.52 kW kg ⁻¹)	10 (9.8 Wh kg ⁻¹)	7

@CNT				
A-Ni-	1 M Na ₂ SO ₄	27.8 (0.49 kW kg ⁻¹)	11 (9 Wh kg ⁻¹)	57
MnBMO//FCNT				
Cu _{0.82} Co _{0.18} HCF//	0.5 M H ₂ SO ₄	36 (1 kW kg ⁻¹)	21 (21 kW kg ⁻¹)	This
h -MoO ₃				work

Comments 4: The characterization and analyses of control sample MoO₃ are insufficient. For example, it has been reported that the structured water in MoO₃ is beneficial for proton migration. However, the manuscript only detected the water content of *h*-MoO₃ via TG curve. Did α -MoO₃ also contain structured water? In my opinion, the manuscript should also supplement detection on the structured water in α -MoO₃.

Our response: Thanks for the good question. We have added more characterization and analyses of α -MoO₃, such as Raman spectra and FTIR spectra in Supplementary Fig. 1 and TG curves in Supplementary Fig. 5, highlighted in yellow. In contrast to *h*-MoO₃, the FTIR and Raman spectra of α -MoO₃ show remarkable change, indicating the different crystal arrangement of the MoO₆ octahedra in α -MoO₃ (Supplementary Figure 1a and b). The absence of the characteristic peaks for O-H illustrates the absence of structural water, which is attributed to the narrow lattice space in α -MoO₃. The TG curve shows the ignorable mass loss in α -MoO₃ with increasing temperature, suggesting that there is no crystal water in α -MoO₃. The structural evolution and degradation mechanism of α -MoO₃ electrodes is evaluated using *in situ* XRD measurements. During the initial discharge, the new diffraction peaks at 50–55° appear, while some diffraction peaks (such as (110)) almost disappear, suggesting the formation of a new phase H_xMoO₃ (Supplementary Fig. 22a-c). During charging and the following discharging, a typical new/old phase transition reaction occurs, accompanying the intercalation/deintercalation of protons in α -MoO₃ (Supplementary Fig. 22d-e). Particularly, the obvious diffraction peak shift during the electrochemical reaction is

associated with the strong electrostatic interaction between charge carriers and electrode hosts, leading to huge internal strain and the structure collapse of electrodes.

Supplementary Fig. 1 a Raman spectra of α -MoO₃. b FTIR spectra of α -MoO₃.

Supplementary Fig. 5 TG curves of α -MoO₃ nanorods.

Supplementary Fig.22 Structural evolution of α -MoO₃. **a** The first discharge curve of α -MoO₃ (a current density of 0.03 A g⁻¹). **b** Contour plots of *in situ* XRD during the first discharge process. **c** *In situ* XRD patterns of different diffraction peaks during the first discharge. **d** GCD curves of α -MoO₃ (the first charge and second discharge processes). **e** *In situ* XRD patterns during charge and discharge processes. **f** *In situ* XRD pattern of different diffraction peak during charge and discharge processes.

Comments 5: More details on theoretical calculations should be supplemented. The manuscript calculated the proton transfer behavior in both *h*-MoO₃ and α -MoO₃. According to the computational details, the manuscript seems to only describe the model construction of *h*-MoO₃. What about α -MoO₃? Corresponding details should be supplemented.

Our response: Thanks for the good question. The model construction of α -MoO₃ has been added in Computational parameters and models. For α -MoO₃, the orthorhombic cell with space group Pbnm was used to optimize both the lattice and ion positions. To calculate crystal structure and diffusion energy barrier, supercells containing 2×1×2 conventional cells were used, with the Monkhorst–Pack scheme *k*-point grid set to 5×3×5.

Comments 6: For the structural model of $\text{MoO}_3 \cdot 0.7\text{H}_2\text{O}$ used in theoretical calculations, the author put four H_2O molecules into MoO_3 unit cell. Is the constructed model consistent with the actual crystal model of $\text{MoO}_3 \cdot 0.7\text{H}_2\text{O}$? The water molecules in $\text{MoO}_3 \cdot 0.7\text{H}_2\text{O}$ are crystal water molecules owning strong chemisorption interaction with MoO_3 crystal. However, the water molecules in theoretical model seem to only physically adsorb around MoO_3 unit according to Fig. 3. To confirm the reliability of constructed model, the author should compare its information on lattice structure (such as lattice parameter after water adsorption) with actual crystal structure.

Our response: Thanks for the good question. The structural model of $\text{MoO}_3 \cdot 0.7\text{H}_2\text{O}$ was built and optimized based on physical characterizations and density functional theory (DFT). To show the proton transport and storage process more clearly, a local image of internal tunnels of the actual crystal model $\text{MoO}_3 \cdot 0.7\text{H}_2\text{O}$ is provided in Fig. 3. Hexagonal MoO_3 has large one-dimensional hexagonal tunnels (the internal channel diameter is ca 0.94 nm) along the c-axis that are formed through interconnecting MoO_6 octahedra, giving additional physical space to the confined fluids such as water molecules to assist ions to achieve a non-diffusion limited transport mechanism. Since water molecules are physically confined in the hexagonal framework, the escape of water molecules does not cause lattice disruption, but only changes some physical characteristics. XRD patterns and lattice constants of as-synthesized and annealed *h*- MoO_3 at 350 °C show the negligible change of lattice structure (Fig. R3 and Table R1), which confirms the reliability of constructed model. Similar physically adsorbed water molecules are found in other materials, such as tunnel water molecules in $\text{WO}_3 \cdot n\text{H}_2\text{O}$ (Nano Lett. 2015, 15, 6802, J. Phys. Chem. C 2021, 125, 11508) and zeolite water in PBAs (Nat. Energy 2019, 4, 123) and MOFs (Chem. Sci., 2019, 10, 16).

Fig. R3 XRD patterns of as-synthesized h -MoO₃ and annealed h -MoO₃ at 350 °C.

Table R1 Lattice constants of as-synthesized h -MoO₃ and annealed h -MoO₃ at 350 °C.

Lattice constants	a (Å)	c (Å)	V (Å ³)
as-synthesized h -MoO ₃	10.53	14.88	1429
annealed h -MoO ₃	10.55	14.89	1435

Comments 7: According to the discussion in section Structural evolution of h -MoO₃, the proton intercalation and storage behavior are related to the exerted potential. According to the description of computational details, the AIMD simulation did not consider the electrochemical environment. In this case, can the performed AIMD simulation accurately describe the reaction and structure evolution process during charge and discharge? The author should supplement essential explanation for this point.

Our response: Thanks for the good question. MD simulations in different electrochemical environment were performed to describe the reaction and structure evolution process, where H_{0.5}MoO₃, HMoO₃, and H₂MoO₃ (different H contents correspond to different potentials) occur in electrochemical potential stage I, II, and III in Fig 4a, respectively (Supplementary Fig. 21 and Fig. 5b). Based on AIMD simulation, protons are intercalated into the electrode bulk phase to generate successively three compositions during the first discharge process. In stage I, protons will combine with crystal waters to form hydronium—H₃O⁺. In stage II, H bonds further with H to form

H-H and fills in hexagonal tunnels. In stage III, oxygen atoms of the MoO₆ octahedra can be occupied, accompanied by H bonding to O as O-H to form H_xMoO₃ (Fig. 5a). The binding energy of formed H₂ and H₃O⁺ is too high to kick out free protons in the subsequent charging process (Fig. 5c), leading to 1.34 mol hydrogen ions permanently occupying the internal space of *h*-H_{1.34}MoO₃·0.7H₂O. *h*-H₂MoO₃·0.7H₂O undergoes a structural self-optimization process to generate 0.66 mol protons acted as charge carriers. Then after the electrode is charged to 0 V, the protons binding with H_xMoO₃ (about 0.66 mol protons) can be de-intercalated from *h*-H₂MoO₃·0.7H₂O to form *h*-H_{1.34}MoO₃·0.7H₂O, and are reversibly inserted/de-inserted during following charging/discharging reaction to achieve outstanding structural stability. The first discharge process is as follows:

In the subsequent charging and discharging process, the reaction mechanism is described according to the following equation:

Supplementary Fig. 21 The distribution of H coordination environments from the room temperature AIMD simulations of *h*-H_{0.5}MoO₃·0.7H₂O (a) and *h*-HMoO₃·0.7H₂O (b) structure.

Fig. 5 Energy storage mechanism of h -MoO₃. **a** Schematic depiction of the energy storage mechanism in h -MoO₃·0.7H₂O. **b** The distribution of H coordination environments from the room temperature AIMD simulations of h -H₂MoO₃·0.7H₂O structure. **c** The formation energies of H vacancy with three different H coordination environments of h -H₂MoO₃·0.7H₂O structure.

Comments 8: There are some typos in the manuscript. For example, “ α -MoO₃” in line 138 should be corrected as “ α -MoO₃”. The author should check the manuscript more carefully.

Our response: Thanks for the good question. “ α -MoO₃” in line 138 has been corrected as “ α -MoO₃”, and other typos have been corrected in the manuscript.

REVIEWERS' COMMENTS

Reviewer #2 (Remarks to the Author):

Unfortunately, the reviewer could not see meaningful differences from previous works (References 25 and 26) in that they have micrometer dimensions (2-4 micrometers in diameter and 10-30 micrometers in length) from micrometer-scale rods reported in this work. More concerns also exist as the authors try to compare the volumetric performances between previous nanostructures and this work by excluding other microstructures such as References 25 and 26. Besides, in Supplementary Table 1, it might be better for the authors to use 1M H₂SO₄ rather than 0.5M H₂SO₄ for comparison. The specific capacitance of 569 F/g (0.5 M H₂SO₄) is not really impressive. Indeed, the maximum energy density (36 Wh/kg at 1 kW/kg) in Supplementary Table 4 is not impressive if one could see great progress in the specific energy density close to 150 Wh/kg of hybrid capacitors, thereby far exceeding those of this work.

For the mass loadings with 3, 5, and 10mg/cm², Figure S10b shows that the areal capacitances with 10mg/cm² at various scan rates are larger than those with 3mg/cm². I consider that it should be a mistake. Also, comparison with those of a very porous activated carbon might be not reasonable. It would be better to compare those of high-density carbonized MOFs.

GGA's are inaccurate in determining exchange energy, but they use the exact kinetic operator, contrary to the comments from the authors. In addition, the authors did not address which correlation functional was used in DFT calculations.

In summary, although the authors made great efforts to improve the quality of the paper, it is considered that this work suffers from uniqueness and novelty.

Reviewer #3 (Remarks to the Author):

I think the authors have addressed the questions or concerns raised by the reviewers. The current manuscript can be accepted for publication.

We thank the reviewers for their time and very useful comments in improving the quality of this manuscript. Provided below is our detailed response to each question.

REVIEWERS' COMMENTS

Reviewer #2 (Remarks to the Author):

Comments 1: Unfortunately, the reviewer could not see meaningful differences from previous works (References 25 and 26) in that they have micrometer dimensions (2-4 micrometers in diameter and 10-30 micrometers in length) from micrometer-scale rods reported in this work. More concerns also exist as the authors try to compare the volumetric performances between previous nanostructures and this work by excluding other microstructures such as References 25 and 26. Besides, in Supplementary Table 1, it might be better for the authors to use 1 M H₂SO₄ rather than 0.5 M H₂SO₄ for comparison. The specific capacitance of 569 F/g (0.5 M H₂SO₄) is not really impressive. Indeed, the maximum energy density (36 Wh/kg at 1 kW/kg) in Supplementary Table 4 is not impressive if one could see great progress in the specific energy density close to 150 Wh/kg of hybrid capacitors, thereby far exceeding those of this work.

Our response: Thanks for the good question.

(1) Differing from the previous studies on photochromic and electrochromic properties of nanometer-scale *h*-MoO₃ in reference 75 and ammonium storage properties of nanometer-scale *h*-MoO₃ in reference 25, we report for the first time the study on proton storage properties of micrometer-scale bulk *h*-MoO₃. Specifically, the *h*-MoO₃ electrodes with micrometer-sized superstructure motifs deliver an ultra-high volumetric capacitance and ultra-stable cycle life. As far as we know, there are few reported pseudocapacitive materials achieving excellent electrochemical performance in such large particle sizes.

(2) We have tried our best to compare the volumetric performance of micrometer-sized *h*-MoO₃ with advanced electrodes, which surpasses most electrode materials previously

reported, such as two-dimensional transition metal carbides/nitrides (MXenes), metal oxides, metal sulfides, metal-organic frameworks (MOFs), redox Graphene (rGO), conductive polymers, and activated carbon. We would like to request the reviewers to review carefully this paper, where we have not artificially excluded other micrometer-scale electrode materials from the comparison (it is different from the previous studies on photochromic and electrochromic properties of nanometer-scale h -MoO₃ in reference 75 and ammonium storage properties of nanometer-scale h -MoO₃ in reference 25).

(3) The electrochemical performance of h -MoO₃ electrodes was measured in 1 M and 0.5 M H₂SO₄, in which the similar electrochemical behavior and performance is shown (Fig. R1 and Fig. 2a). The H₂SO₄ electrolyte with higher concentration might lead to the material dissolution and current collector corrosion. To ensure the cycle life, safety, cost, and environmental friendliness, we performed all the electrochemical performance tests of h -MoO₃ electrodes using 0.5 M H₂SO₄. As shown in Supplementary Table 1, the micrometer-sized bulk electrode material with a specific capacitance of 569 F g⁻¹ is exceptional in comparison to other electrodes.

Figure R1 Capacitance of h -MoO₃ electrodes at various scan rates in 1 M H₂SO₄.

(4) Taking advantage of the high capacity and rapid kinetics of proton topochemistry in bulk h -MoO₃ engenders an exciting avenue for breaking through the power limit of batteries and the energy limit of capacitors. The proton pseudocapacitor remains a high energy density of 21 Wh kg⁻¹ at a maximum power density of 21 kW kg⁻¹, which will

maximize the energy and power density of supercapacitors. The electrochemical performance of this proton pseudocapacitor (especially the energy density at maximum power density) surpasses not only most supercapacitors in acid electrolytes, but also many typical supercapacitors in alkaline and neutral electrolytes (Supplementary Table 4), such as 2-GCE//AC (23.4 Wh kg⁻¹ at 3.9 kW kg⁻¹ and 21.1 kW kg⁻¹ at 10 Wh kg⁻¹) and 1@Ti₃C₂T_x//ACL (32.2 Wh kg⁻¹ at 2.4 kW kg⁻¹ and 12 kW kg⁻¹ at 19.7 Wh kg⁻¹) in acid electrolytes, MLMO//MLMO (34.1 Wh kg⁻¹ at 0.8 kW kg⁻¹ and 14.5 kW kg⁻¹ at 19.4 Wh kg⁻¹) in alkaline electrolytes, and MoO₃@CNT//MnO₂@CNT (27.8 Wh kg⁻¹ at 0.52 kW kg⁻¹ and 10 kW kg⁻¹ at 9.8 Wh kg⁻¹) in neutral electrolytes.

Supplementary Table 4. Electrochemical performance comparison of our work with various supercapacitors reported in the last five years.

Electrode	Electrode loading (mg)	Electrolyte	Max Energy density (Wh kg ⁻¹)	Max power density (kW kg ⁻¹)	Ref.
2-GCE//AC	3:1	3 M H ₂ SO ₄	23.4 (3.9 kW kg ⁻¹)	21.1 (10 Wh kg ⁻¹)	47
RuO ₂ /Hex-Aza-COF-3	9.6:8	1 M H ₂ SO ₄	23.3 (0.67 kW kg ⁻¹)	—	48
1@Ti ₃ C ₂ T _x //ACL	1:0.72	2 M H ₃ PO ₄	32.2 (2.4 kW kg ⁻¹)	12 (19.7 Wh kg ⁻¹)	49
W ₁₈ O ₄₉ /Ti ₃ C ₂ T _x //RuO ₂ @CC	2.4:1.2	1 M H ₂ SO ₄	29.6 (0.75 kW kg ⁻¹)	7 (24.1 Wh kg ⁻¹)	50
MWCNT/RuO ₂ //W	—	1 M H ₂ SO ₄	27.2 (0.75 kW kg ⁻¹)	—	51
O ₃ /Ti ₃ C ₂ T _x					
AC//2-CPE	—	0.5 M H ₂ SO ₄	16.1 (1.7 kW kg ⁻¹)	10 (10.8 Wh kg ⁻¹)	52

PYT/GN 4-5//A- Ti ₃ C ₂ T _x	1.7:1.5	1 M H ₂ SO ₄	18.4 (0.7 kW kg ⁻¹)	7 (13.2 Wh kg ⁻¹)	53
Ni-S/1d-Ti ₃ C ₂ //d- Ti ₃ C ₂	2.1:1.3	6 M KOH	20 (0.5 kW kg ⁻¹)	10 (5.1 Wh kg ⁻¹)	54
STO//STO	0.45:0.45	3 M KOH	27.8 (0.3 kW kg ⁻¹)	19.2 (—)	55
MLMO//MLMO	1:1	3 M KOH	34.1 (0.8 kW kg ⁻¹)	14.5 (19.4 Wh kg ⁻¹)	56
MnO ₂ @CNT// MoO ₃ @CNT	1.3:1	1 M Na ₂ SO ₄	27.8 (0.52 kW kg ⁻¹)	10 (9.8 Wh kg ⁻¹)	7
A-Ni- MnBMO//FCNT	0.5:1	1 M Na ₂ SO ₄	27.8 (0.49 kW kg ⁻¹)	11 (9 Wh kg ⁻¹)	57
Cu _{0.82} Co _{0.18} HCF// h -MoO ₃	1.2:1	0.5 M H ₂ SO ₄	36 (1 kW kg ⁻¹)	21 (21 kW kg ⁻¹)	This work

Comments 2: For the mass loadings with 3, 5, and 10 mg cm⁻², Figure S10b shows that the areal capacitances with 10 mg cm⁻² at various scan rates are larger than those with 3 mg cm⁻². I consider that it should be a mistake. Also, comparison with those of a very porous activated carbon might be not reasonable. It would be better to compare those of high-density carbonized MOFs.

Our response: Thanks for the good question. As the mass loading of electrodes increases, the area specific capacitance gradually increases within a certain mass range. It is reasonable that the areal capacitances with 10 mg cm⁻² are larger than those with 3 mg cm⁻² at various scan rates, the similar results were found in other materials (Nat. Energy 2018, 3, 30; Energy Environ. Sci. 2020, 13, 949; Adv. Mater. 2020, 32, 1906652; J. Mater. Chem. A 2020, 8, 18933). Indeed, due to the poor kinetics of thick electrodes,

the phenomenon mentioned by the reviewer may occur at a higher scan rate. The low synthesis efficiency and high cost of carbonized MOFs hinder the practical application in electrochemical energy storage. To meet the need for practical applications, it is reasonable that the areal capacitance of $h\text{-MoO}_3$ has been compared with commercialized porous activated carbon (Nat. Nanotechnol. 2022, 17, 153; Energy Stor. Mater. 2020, 28, 307). Following the reviewer's suggestion, we have removed the areal capacitance comparison of $h\text{-MoO}_3$ and porous activated carbon in the **Supplementary Fig. 15** and **Revised Manuscript**.

Supplementary Fig. 15 a Volumetric capacitance of $h\text{-MoO}_3$ electrodes at different mass loadings. **b** Areal capacitance of $h\text{-MoO}_3$ electrodes at different mass loadings.

Comments 3: GGAs are inaccurate in determining exchange energy, but they use the exact kinetic operator, contrary to the comments from the authors. In addition, the authors did not address which correlation functional was used in DFT calculations.

Our response: Thanks for the good question. I apologize for the confusion in my previous response. You are correct that GGAs, including the PBE functional, use the exact kinetic operator in their formulations. Therefore, they are generally accurate in determining the kinetic energy. The inaccuracy I mentioned earlier primarily refers to the exchange energy, not the kinetic energy. GGAs can underestimate the exchange

energy, especially in systems with strong correlation effects or when long-range interactions are important. This underestimation can lead to errors in predicting certain properties, such as the energetics of reactions or the electronic structure of materials. While GGAs, including PBE, provide reasonable descriptions of the kinetic energy, they may not fully capture the non-local nature of the exchange interaction, which can affect the accuracy of certain properties. To improve the description of exchange energy, more advanced functionals, such as hybrid functionals or meta-GGAs, can be employed. In this paper, meta-GGA was used to optimize and verify the GGA results, suggesting the applicability of meta-GGA and GGA in this system (J. Chem. Phys. 2000, 112, 2643; Nat. Chem. 2016, 8, 831; J. Chem. Phys. 2020, 152, 224101). The GGA and meta-GGA exchange functional was widely adopted for structural relaxations and energy calculations (Nature 2020, 585, 63; Nature 2022, 606, 305; Nat. Energy 2021, 6, 706; Nat. Mater. 2023, 22, 353; Nat. Energy 2019, 4, 123; Sci. Adv. 2023, 9, 4589; NPI Comput. Mater. 2018, 4, 60).

Additionally, all calculations were carried out by using the projector augmented wave method in the framework of the density functional theory (DFT), as implemented in the Vienna ab-initio Simulation Package (VASP). The generalized gradient approximation (GGA) functional, Perdew–Burke–Ernzerhof (PBE) exchange–correlation functional, and projector-augmented wave (PAW) were used.

Reviewer #3 (Remarks to the Author):

Comments: I think the authors have addressed the questions or concerns raised by the reviewers. The current manuscript can be accepted for publication.

Our response: The authors thank the reviewer for his/her recommendation for publication in Nature Communications.